# A standardized methodology for the validation of air quality forecast applications (F-MQO): Lessons learnt from its application across Europe

Lina Vitali[1], Kees Cuvelier[2,a], Antonio Piersanti[1], Alexandra Monteiro[3], Mario Adani[1], Roberta Amorati[4], Agnieszka Bartocha[5], Alessandro D'Ausilio[6], Paweł Durka[7], Carla Gama[3], Giulia Giovannini[4], Stijn Janssen[6], Tomasz Przybyła[5], Michele Stortini[4], Stijn Vranckx[6], Philippe Thunis[2]

[1]National Agency for New Technologies, Energy and Sustainable Economic Development (ENEA), Department for Sustainability, Bologna, Italy
[2]European Commission - Joint Research Centre (JRC), Ispra, Italy
[3]CESAM, Department of Environment, University of Aveiro, Aveiro, Portugal
[4]Regional Agency for Prevention Environment and Energy (ARPAE) of the Emilia-Romagna Region, Bologna, Italy
[5]ATMOTERM, Opole, Poland
[6]Flemish Institute for Technological Research (VITO), Mol, Belgium
[7]Institute of Environmental Protection (IEP) - National Research Institute, Warsaw, Poland

[a]Retired with Active Senior Agreement

*Correspondence to*: Philippe Thunis (Philippe.THUNIS@ec.europa.eu)

**Abstract.** A standardized methodology for the validation of short-term air quality forecast applications was developed in the framework of FAIRMODE activities. The proposed approach, focusing on specific features to be checked when evaluating a forecasting application, investigates the model capability to detect sudden changes of pollutants concentrations levels, to predict threshold exceedances and to reproduce air quality indices. The proposed formulation relies on the definition of specific forecast Modelling Quality Objective and Performance Criteria, defining the minimum level of quality to be achieved by a forecasting application when it is used for policy purposes. The persistence model, which uses the most recent observed value as predicted value, is used as benchmark for the forecast evaluation. The validation protocol has been applied to several forecasting applications across Europe, using different modelling paradigms and covering a range of geographical contexts and spatial scales. The method is successful, with room for improvement, in highlighting shortcomings and strengths of forecasting applications. This provides a useful basis for using short-term air quality forecast as a supporting tool for correct information to citizens and regulators.

## 1 Introduction

Air pollution models play a key role in both enhancing the scientific understanding of atmospheric processes and supporting policy in adopting decisions aimed at reducing human exposure to air pollution. Current European Air Quality Directives

(AQD), 2008/50/EC (European Union, 2008) and 2004/107/EC (European Union, 2004), and even more the proposal of their revision (European Union, 2022), encourage the use of models in combination with monitoring in a wide range of applications. Indeed, models have the advantages of being cheaper than measurements and covering continuously and
simultaneously large areas. Advances in the knowledge of atmospheric processes and the enhancement in computational technologies fostered the usage of 3-dimensional numerical models, the Chemical Transport Models, not only for air quality assessment (retrospective simulation of historical air quality scenarios in support of regulation and planning) but also for real-time air quality forecasting. Indeed, during last decades, air quality forecasting systems based on Chemical Transport Models have been rapidly developed and they are currently operational in many countries, providing early air quality
warnings that allow policy makers and citizens to take measures in order to reduce human exposure to unhealthy levels of air pollution. On European scale, a real-time air quality forecasting system (Marécal et al., 2015) is operational since 2015 in the framework of the Copernicus Atmosphere Monitoring Service (CAMS) and currently includes eleven numerical air quality models, contributing to the CAMS Regional Ensemble production ([https://regional.atmosphere.copernicus.eu/](https://regional.atmosphere.copernicus.eu/)). Several review papers are available in literature, comprehensively describing current status and emerging challenges in real-time air
quality forecasting  (e.g. Kukkonen et al., 2012; Zhang et al., 2012; Baklanov et al., 2014; Ryan, 2016; Bai et al., 2018; Baklanov and Zhang, 2020; Sokhi et al., 2022), including air quality forecasting system based on artificial intelligence methods (e.g. Cabaneros et al., 2019; Masood and Ahmad, 2021; Zhang et al., 2022).

A thorough assessment of model performances is fundamental to build confidence in models' capabilities and potentials and becomes imperative when model applications support policymaking. Moreover, performance evaluation is very important
also for research purposes, since investigating models' strengths and limitations provides essential insights for planning new model developments.

The main goal of a model evaluation process is to prove that the performances are satisfactory for its intended use, in other words, that it is "fit for purpose" (e.g. Hanna and Chang, 2012; Dennis et al., 2010; Baklanov et al., 2014; Olesen, 1996). Indeed, to be able to determine whether a model application is "fit for purpose", its purpose should be stated at the outset.
Since air quality models are used to perform various tasks (e.g. assessment, forecasting, planning), depending on the aim pursued, different evaluation strategies should be put into practice.

Several scientific studies have already proposed different evaluation protocols or suggested recommendations for good practices (e.g. Seigneur et al., 2000; Chang and Hanna, 2004; Borrego et al., 2008; Dennis et al., 2010; Baklanov et al., 2014; Emery et al., 2017). Models applied for regulatory air quality assessment are commonly evaluated by statistical analysis,
examining how well they match the observations. From literature review, many statistical measures are used to quantify the different aspects of the agreement between simulations and observations. Indeed, no single metric is likely to reveal all aspects of model skills. So, the usage of several metrics, in concert, is generally recommended to support an in-depth assessment of performances. Zhang et al. (2012) provide an exhaustive collection of the most used metrics. The list includes both traditional discrete statistical measures (e.g. Emery et al., 2017), quantifying the differences between modelled and

observed values, and categorical indices (e.g. Kang et al., 2005), describing the capability of the model application in predicting categorical answers (e.g. exceedances of limit values).

Ideally, a set of performances criteria should be given within a model evaluation exercise, stating if the model application skills can be considered adequate. As an example, Boylan and Russell (2006) and Chemel et al. (2010) proposed performance criteria and goals for mean fractional bias (*MFB*) and mean fractional error (*MFE*) concerning the validation of

aerosol and ozone modelling applications, respectively. More in details, criteria define the acceptable accuracy level whereas goals specify the highest expected accuracy. Russell and Dennis (2000), citing Tesche et al. (1990), provided informal fitness criteria for urban photochemical modelling, according to some commonly used metrics (i.e. normalized bias, normalized gross error, unpaired peak prediction accuracy). Indeed, these recommendations are based on outcomes of performances skills from previous model studies. Specifically concerning air quality forecasting, in the framework of CAMS

Regional Ensemble production, performances targets (Key Performance Indicators, KPI) are defined for the root mean square error (RMSE) in simulating ozone, nitrogen dioxide and aerosol. Their compliance is regularly reported within the Quarterly Evaluation and Quality Control Reports (https://atmosphere.copernicus.eu/regional-services).

Concerning both the definition of protocols for model evaluation and the proposal of performances criteria, an important contribution came in the last decades from the activities and the coordination efforts of the Forum for Air quality Modeling

in Europe (FAIRMODE, https://fairmode.jrc.ec.europa.eu/home/index). FAIRMODE was launched in 2007 as a joint initiative of the European Environment Agency (EEA) and the European Commission Joint Research Centre. Its primary aim is to promote the exchange of good practices among air quality modellers and users and foster harmonization in the use of models by European Member States, with emphasis on model application under the European Air Quality Directives. In this context, one of the main activities of FAIRMODE has been the development of harmonized protocols for the validation and

the benchmarking of modelling applications. These protocols include the definition of common standardized Modelling Quality Objectives (*MQO*) and Modelling Performance Criteria (*MPC*) to be fulfilled in order to ensure a sufficient level of quality of a given modelling application. More in details, an evaluation protocol was proposed for the evaluation of model applications for regulatory air quality assessment. The methodology (Thunis et al., 2012b; Pernigotti et al., 2013; Thunis et al., 2013; Janssen and Thunis, 2022) is based on the comparison of model-observation differences (namely, the root mean

square error) with a quantity proportional to the measurement uncertainty. The rationale is that a model application can be considered acceptable if the model-measurement differences remain within a given proportion of the measurement uncertainty. The approach is consolidated in the DELTA Tool software (Thunis et al. 2012a, https://aqm.jrc.ec.europa.eu//Section/Assessment/Download). It has reached a good level of maturity and it has been widely used and tested by model developers and users (Georgieva et al., 2015; Carnevale et al., 2015; Monteiro et al., 2018; Kushta

et al., 2019). This approach focuses on applications related to air quality assessment, in the context of the AQD 2008/50/EC (European Union 2008), taking into account pollutants and metrics consistently with the AQD requirements.

Recently, FAIRMODE worked on developing and testing additional quality control indicators to be complied when evaluating a forecast application, extending the approach used for assessment applications. A scientific consensus was

reached, focusing on the model ability in the specific purpose of accurately predicting sudden changes and peaks in the

pollutant concentration levels. The proposed methodology, based on the usage of the persistence model (e.g. Mittermaier, 2008) as a benchmark, is now publicly available for testing and application.

This paper describes this new standardized approach and is organized as follows. Sect. 2 illustrates the rationale and the main features of the developed methodology. Sect. 3 describes the setup of the forecasting simulations to which the methodology was applied, including information on the monitoring data used for the validation. Results are presented in

Sect. 4, focusing on lessons learnt from the application of the proposed approach in different geographical contexts and spatial scales. Finally, some conclusions are drawn in Sect. 5 together with hints for further developments.

## 2 Methodology

The validation protocol proposed in this work is specific for forecasting evaluation. It is an extension of the consolidated and well documented methodology fostered by FAIRMODE for the evaluation of model applications for regulatory air quality

assessment. Therefore, it is recommended that the metrics suggested when evaluating forecasting applications are applied in addition to the standard assessment *MQO*, as defined in Janssen and Thunis (2022). This section describes the main features of the proposed protocol, which focuses on the model capability to (1) detect sudden changes of concentrations levels (Sect. 2.1), (2) predict threshold exceedances (Sect. 2.2) and (3) reproduce air quality indices (Sect. 2.3). Note that the proposed approach is not exhaustive. It does not evaluate all relevant features of a forecast application and other analyses will be

helpful to gain further insights into the behaviour, the strengths and the shortcomings of a forecast application.

The methodology, as currently implemented in the DELTA Tool software, supports the following pollutants and time averages: $NO_2$ daily maximum, $O_3$ daily maximum of 8-hour average, PM10 and PM2.5 daily mean.

### 2.1 Forecast Modelling Quality Objective (*MQO_f*) based on the comparison with the persistence model

Predicting the status of air quality is useful in order to prevent or reduce health impacts from acute episodes and to trigger

short-term action plans. Therefore, it is of main interest to verify the forecast applications ability in getting the purpose of accurately reproducing sudden changes in the pollutant concentration levels. To account for this, within the proposed protocol, the main evaluation assessment of the "fitness for purpose" of a forecast application is based on the usage, as a benchmark, of the persistence model, that is by default not able to capture any changes in the concentration levels, since measurement data of the previous day are used as an estimate for the full forecast horizon. Indeed, the persistence approach

is the simplest method for predicting the future behaviour, if no other information is available and is often used as a reference in verifying the performances of weather forecasts (e.g. Knaff and Landsea, 1997; Mittermaier, 2008).

Within the proposed forecasting evaluation protocol, the root mean square error of the forecast model is compared with the root mean square error of the persistence model. More in detail, a forecast Modelling Quality Indicator (*MQI_f*) is defined as the ratio between the two *RMSEs*, i.e.

$$MQI_f = \sqrt{\frac{\frac{1}{N}\sum_{i=1}^{N}(M_i - O_i)^2}{\frac{1}{N}\sum_{i=1}^{N}(P_i - O_i)^2}}$$ (1)

where $M_i$, $P_i$, $O_i$ represent respectively the forecast, the persistence, and the measured values for day $i$, and $N$ is the number of days included in the time series.

The persistence model uses the observations from the previous day as an estimate for all forecast days. As an example, we can consider a 3 day-forecast, providing today (day0), tomorrow (day1), and the day after tomorrow (day2) concentration values. If today is 5th February, persistence model uses data referring to yesterday (4th February) for all forecast data produced today. So, $P_i$ refers to $O_{i-1}$ for day0 (5th February), it refers to $O_{i-2}$ for day1 (6th February) and it refers to $O_{i-3}$ for day2 (7th February). More generally, the persistence model is related to the forecast horizon ($FH =0, 1, 2$, etc.) as follows:

$$P_i = O_{i-1-FH} \pm U(O_{i-1-FH})$$ (2)

where the measurement uncertainty U is also taken into account, consistently with the FAIRMODE approach. The methodology for estimating the measurement uncertainty as a function of the concentrations values is described in Janssen and Thunis (2022), where the parameters for its calculation for PM, $NO_2$ and $O_3$ are provided as well. It is important to note that we use as representative for the measurement uncertainty the 95[th] percentile highest value among all uncertainty values. For PM10 and PM2.5 the results of a JRC instrument inter-comparison (Pernigotti et al., 2013) have been used whereas a set of EU AIRBASE stations available for a series of meteorological years has been used for $NO_2$ and analytical relationships have been used for $O_3$. These 95[th] percentile uncertainties only include the instrumental error. More details are provided in Appendix A. The fulfilment of the forecast Modelling Quality Objective ($MQO_f$) is proposed as a necessary but not sufficient quality test to be achieved by the forecasting application. The $MQO_f$ is fulfilled when $MQI_f$ is less than or equal to 1, indicating that the forecast model performs better (within the measurement uncertainty) than the persistence one, with respect to its capability of detecting sudden changes of concentrations levels.

Within the proposed protocol, two aspects are included in a single metric ($MQI_f$ ): (1) check how well the model prediction compares with measurements and (2) check whether the model prediction performs better than a given benchmark (here the persistence model).

The magnitude of the $MQI_f$ score, since it is referenced to a benchmark, is dependent on the skill of the benchmark itself. To account for this, additional Modelling Performance Indicators ($MPIs$) are proposed as part of the evaluation protocol, based on the mean fractional error ($MFE$), a normalized statistical indicator widely used in literature, defined as follows:

$$MFE = \frac{2}{N}\sum_{i=1}^{N}\frac{|M_i - O_i|}{(M_i + O_i)}$$ (3)

Based on this indicator, two different $MPIs$ are defined and both included within the protocol: 1) $MPI_1= MFE_f/MFE_p$ that compares the forecast model performances with the persistence model ones; 2) $MPI_2= MFE_f/MF_U$ that evaluates forecast

performances regardless of persistence aspects, using an acceptability threshold based on measurement uncertainty, where $MF_U$ is the Mean Fractional Uncertainty, defined as follows:

$$MF_U = \frac{1}{N} \sum_{i=1}^{N} \frac{2U(O_i)}{O_i} \tag{4}$$

Using the uncertainty parameters provided in Table A1 in Appendix A, it turns out that $2U(O_i)/O_i$ shows larger values in the low concentration range and then tends towards a constant (0.5 for $NO_2$, 0.3 for $O_3$, 0.55 for PM10, 0.6 for PM2.5) at higher concentration values (Fig. A1, in Appendix A). So, the choice of $MF_U$ as the acceptability threshold is consistent with performances criteria and goals defined in literature for PM (Boylan and Russell, 2006) and $O_3$ (Chemel et al., 2010) and it has the advantage that it does not introduce any additional free parameters and it can be applied to all pollutants for which uncertainty parameters are set. For both *MPIs*, Modelling Performance Criteria (*MPC*) are proposed, being fulfilled when *MPIs* are less or equal to 1.

**2.2 Assessment of modelling application capability in predicting Threshold Exceedances**

When a forecasting system is used for policy purposes, it is of main interest to verify the skill in predicting categorical answers (yes/no) in relation to exceedances of specific threshold levels, e.g. the limit values set by the current European legislation (European Union, 2008).

To account for this, the most commonly used threshold indicators (as defined in Table 1) are included in the proposed validation approach, based on the 2x2 contingency table (Table B1, in Appendix B) representing the joint distribution of categorical events (below/above the threshold value) predicted by the model and observed by the measurements. Namely, $GA_+$ represents the number of correctly forecasted exceedances, $GA_-$ represents the number of correctly forecasted non-exceedances, *FA* (False Alarms) represents the number of forecasted exceedances that were not observed, and *MA* (Missed Alarms) represents the number of observed exceedances that were not forecasted.

All metrics included are listed in Table 1, ranging from 0 to 1, being 1 the optimal value.

**Table 1.** Categorical metrics included in the validation protocol.

| Metrics | Mathematical Expression |
|---|---|
| Accuracy | $ACC = \dfrac{GA_+ + GA_-}{FA + GA_+ + GA_- + MA}$ |
| Success Ratio | $SR = \dfrac{GA_+}{FA + GA_+}$ |
| Probability of Detection | $PD = \dfrac{GA_+}{GA_+ + MA}$ |

| | |
|---|---|
| FBias score | $$FB = \frac{FA + GA_+}{GA_+ + MA}$$ |
| Threat score | $$TS = \frac{GA_+}{FA + GA_+ + MA}$$ |
| Gilbert Skill score | $$GSS = \frac{GA_+ - H}{FA + GA_+ + MA - H}$$ $$with\ H = \frac{(GA_+ + MA)(FA + GA_+)}{FA + GA_+ + GA_- + MA}$$ |

## 2.3 Assessment of modelling application capability in predicting Air Quality Indices

One of the main objectives of a forecasting system is to provide citizens with simple information about local air quality and its potential impact on their health, with special regard to the sensitive and vulnerable groups (i.e., the very young or old,

asthmatics, etc.). Air Quality Indices (AQI) are designed to provide information on the potential effects of the different pollutants on people's health by means of a classification of concentrations values in terms of qualitative categories.

The AQI outcome is commonly provided by operational forecasting systems, therefore its assessment has been included in the proposed validation approach, by means of a simple multiple thresholds assessment. More in detail, the number of days predicted by the forecast model in each category is compared with the corresponding number of measured days.

Of course, the performance assessment depends on the chosen classification table. In the current approach, several AQI tables are available, namely EEA (https://www.eea.europa.eu/themes/air/air-quality-index/index), United Kingdom (https://uk-air.defra.gov.uk/air-pollution/daqi; https://uk-air.defra.gov.uk/air-pollution/daqi?view=more-info), and USEPA (U.S. Environmental Protection Agency, https://www.airnow.gov/aqi/aqi-basics/, Eder et al. 2010) classification tables.

## 3 Forecasting applications: models, setup and monitoring data for validation

The proposed methodology was applied across Europe to evaluate the performances of several forecasting applications. This paper focuses on lessons learnt by the validation of five forecasting applications, based on various methods (both in terms of chemical transport models and statistical approaches) and covering different geographical contexts and spatial scales, from very local to European scale. The key features of the forecast applications are summarized in Table 2. Some more details are provided for each of them in the following, along with information on the monitoring data used for the validation.


**Table 2.** Main features of the forecast applications.

| Forecast application Acronym | Operated by | Modelling System | Modelling approach | Time Period | Horizontal Domain & Resolution | Meteo | Emissions | Boundary Conditions | Data Assimilation |
|---|---|---|---|---|---|---|---|---|---|
| **FA1** | ENEA | MINNI | Chemical Transport Model | Year-long Simulation (2018) | Europe (25°W-45°E, 30°N-72°N) Resolution: 0.1° | IFS | CAMS REG (v5.1) | C-IFS | NO |
| **FA2** | CESAM | WRF-CHIMERE | Chemical Transport Model | Year-long Simulation (2021) | Portugal (10.3°W-5.7°W, 36.4°N-42.6°N) Resolution: 0.05° | NCEP/GFS | EMEP/CEIP | GOCART for dust, LMDz-INCA for gaseous and other aerosol species | NO |
| **FA3** | VITO | OPAQ | Neural Networks | Year-long simulation (2022) | Ireland (10.5°W-5.9°W, 51.4°N-55.4°N) Resolution: 3 km | ECMWF | Not applicable | Not applicable | NO |
| **FA4** | ARPAE | NINFA | Chemical Transport Model | Year-long Simulation (2021) | PREPAIR domain (6.25°E-16.75° E, 43.1°N-47.35°N) Resolution: 0.07° x 0.05° | COSMO | EMEP/CEIP, PREPAIR | kAIROS | NO |
| **FA5** | ATMOTERM | CALPUFF | Dispersion Model | July 2020 - September 2022 | Variable spatial grid-size covering Kosovo (1 km/0.5km) Pristine (200m/50m) | WRF | Kosovo emission inventory | CAMS ENSEMBLE | YES |

### 3.1 MINNI simulation over Europe (FA1)

The first forecast application (FA1) was operated by ENEA applying the MINNI Atmospheric Modelling System (Mircea et al., 2014; D'Elia et al., 2021) on a European domain at 0.1° horizontal spatial resolution. FA1 is a year-long simulation, referring to 2018. MINNI, which is operationally providing air quality predictions over an Italian domain since 2017 (Adani et al., 2020, 2022), was recently added to the ensemble of the eleven models contributing to the CAMS Regional Ensemble production. FA1 was carried out during a preliminary benchmark phase, using CAMS input and setup, but it is not an official CAMS product.

Since no data assimilation was applied within FA1, all available data measured at European background monitoring stations and collected by EEA (E1a at https://discomap.eea.europa.eu/map/fme/AirQualityExport.htm) were considered for the validation.

### 3.2 WRF-CHIMERE simulation over Portugal (FA2)

In Portugal, an air quality modelling system based on the WRF version 3 (Skamarock et al., 2008) and the CHIMERE chemical transport model v2016a1 (Menut et al., 2013; Mailler et al., 2017) is being used for forecasting purposes at daily basis since 2007 (Monteiro et al., 2005, 2007a, b). The modelling setup comprises three nested domains covering part of the North Africa and Europe, with horizontal resolutions of 125 km, 25 km and 5 km for the innermost domain covering Portugal. At the boundaries of the outermost domain, the outputs from LMDz-INCA (Szopa et al., 2009) are used for all gaseous and aerosol species, and dust from the GOCART model (Ginoux et al., 2001). The main human activities emissions (traffic, industries and agriculture, among others) are derived based on data from the annual EMEP/CEIP emission database (available at https://www.ceip.at/webdab-emission-database/), following a procedure of spatial and temporal downscaling. Biogenic emissions are computed online using the MEGAN model (Guenther et al., 2006), while dust emission fluxes are calculated using the dust production model proposed by Alfaro and Gomes (2001).

Data from the Air Quality National Monitoring network (https://qualar.apambiente.pt) is used every year to assess the performance of this forecasting modelling system, usually evaluated at annual basis. This comprehends a group of more than 40 background monitoring stations, classified as urban, suburban and rural environment, according to the classification settled by European legislation.

### 3.3 OPAQ simulation over Ireland (FA3)

The OPAQ (Hooyberghs et al., 2005; Agarwal et al., 2020) statistical forecast system has been configured and applied to forecast pollution levels in Ireland by the Irish EPA and VITO. During the configuration stage neural networks are trained at station level with historical observations, ECMWF-ERA5 reanalysis meteorological data and the CAMS air quality

forecasts. The forecasts at station level are interpolated to forecast maps for the whole country using the detrended kriging model RIO (Janssen et al., 2008; Rahman et al., 2023) which is part of the OPAQ system.

In this study, we present the historical validation results of a feed-forward neural network model that uses 2-metre temperature, vertical and horizontal wind velocity component, CAMS PM10 forecasts, and PM10 observations. More than two years of data are used to configure the OPAQ model. Data from October 2019 to June 2022 are used for training. The model is validated on the data for July to December 2022. The testing holdout sample, used to avoid over fitting, covers a timespan of three months from June to September 2019. The model was optimized using the Adamax algorithm (Kingma and Ba, 2014) with 4 hidden layers and 200 units per layer, the activation function uses sigmoid functions while the mean squared error is used as loss function.

### 3.4 NINFA simulation over Po Valley and Slovenia (FA4)

FA4 was operated by ARPAE applying NINFA, his operational Air Quality Model Chain over Po Valley and Slovenia in the framework of Life Ip PREPAIR project (https://www.lifeprepair.eu/; Raffaelli et al., 2020). The model suite includes a Chemical Transport Model, a meteorological model and an emissions pre-processing tool. The chemical transport model is CHIMERE, v2017r3. Emission data cover the Po Valley (Marongiu et al., 2022), Slovenia and the other regions/countries present in the model domain (http://www.lifeprepair.eu/wp-content/uploads/2017/06/Emissions-dataset_final-report.pdf). The meteorological hourly input is provided by COSMO (http://www.cosmo-model.org; Baldauf et al., 2011; Doms and Baldauf, 2018).The boundary conditions are provided by kAIROS (Stortini et al., 2020).

The database of observed data used in this work, was built with the support of PREPAIR partners providing revised validated data for 2021.

### 3.5 CALPUFF simulation over Kosovo (FA5)

FA5 was operated by ATMOTERM Company between July 2020 and September 2022. Analyses were based on data available from Kosovo Air Quality Portal hosted by the Hydrometeorological Institute of Kosovo and Kosovo Open Data Platform (https://airqualitykosova.rks-gov.net/en/; https://opendata.rks-gov.net/en/organization/khmi). Forecast Service was using the following modelling tools: WRF meteorological prognostic model, CAMS ENSEMBLE Eulerian air quality models and CALPUFF Modelling System with 1 km receptor grid covering the Kosovo territory and 0.5 km grid applied in the main Kosovo cities. In addition a high resolution receptor network was created for Pristine, with the basic grid step of 200 m and 50 m along the roads. The system includes an assimilation module implemented at the post-processing stage using available data from all monitoring stations in Kosovo.

## 4 Results, Lessons learnt and Discussion

The proposed evaluation methodology for forecasting comes on top of the consolidated FAIRMODE protocol for assessment. The assessment $MQO$ therefore comes first to provide a preliminary evaluation of the five forecasting applications (see Appendix C). This section focuses on the outcomes of applying the additional forecast objectives and criteria, in particular on the lessons learnt by their application to different geographical contexts and spatial scales, pointing out to the strengths and shortcomings of the approach.

### 4.1 $MQO_f$ skills versus the capability of predicting Threshold Exceedances

Forecast Modelling Quality Objective ($MQO_f$) outcomes are presented here for three forecasting applications, covering different spatial scales, namely FA1 (European scale), FA2 (national scale), and FA4 (regional scale). Along with $MQO_f$ outcomes, the skills of the three modelling applications in predicting threshold exceedances are provided as well. We present outcomes for PM10 daily mean and $O_3$ daily maximum of 8-hour average, since both indicators have a daily limit value set by the current European legislation (European Union, 2008).

Figs. 1-2, Figs. 3-4, and Figs. 5-6 show the outcomes for FA1, FA2, FA4 applications, respectively. PM10 outcomes are provided in Figs. 1, 3, and 5 while Figs. 2, 4, and 6 present the $O_3$ ones . $MQI_f$ values are provided in the Forecast Target Plots (Janssen and Thunis, 2022), at the top of each Figure. Within these plots, $MQI_f$ is represented by the distance between the origin and a given point (for each monitoring station). Values lower than 1 (i.e. within the green circle) indicate better capabilities than the persistence model (within the measurement uncertainty), whereas values larger than 1 indicate poorer performances. Indeed, the green area identifies the fulfilment of the $MQO_f$ at each monitoring stations. The $MQI_f$ associated to the 90[th] percentile worst station is reported in the upper left corner of the plots. This value is used as the main indicator in the proposed benchmarking procedure: its value should be less than or equal to 1 for the fulfilment of the benchmarking requirements. In other words, within the proposed protocol a forecasting application is considered "fit for purpose" if $MQI_f$ is lower than 1 for at least 90% of the available stations. Note that passing the $MQO_f$ test is intended here as a necessary condition for the use of the modelling results but it must not be understood as a sufficient condition that ensures that model results are of sufficient quality.

The outcomes of all categorical metrics included in the validation protocol are provided at the bottom of each Figure, by means of the Forecast Summary P-Normalized Reports. Within these plots, the statistical distribution (5[th], 25[th], 50[th], 75[th], 95[th] percentiles) of the outcomes of all the indicators defined in Sect. 2.2 are summarized and compared with the corresponding outcomes of the persistence model (i.e. the ratios of the skills are considered). Green area indicates that model performs better than the persistence model for that particular indicator.

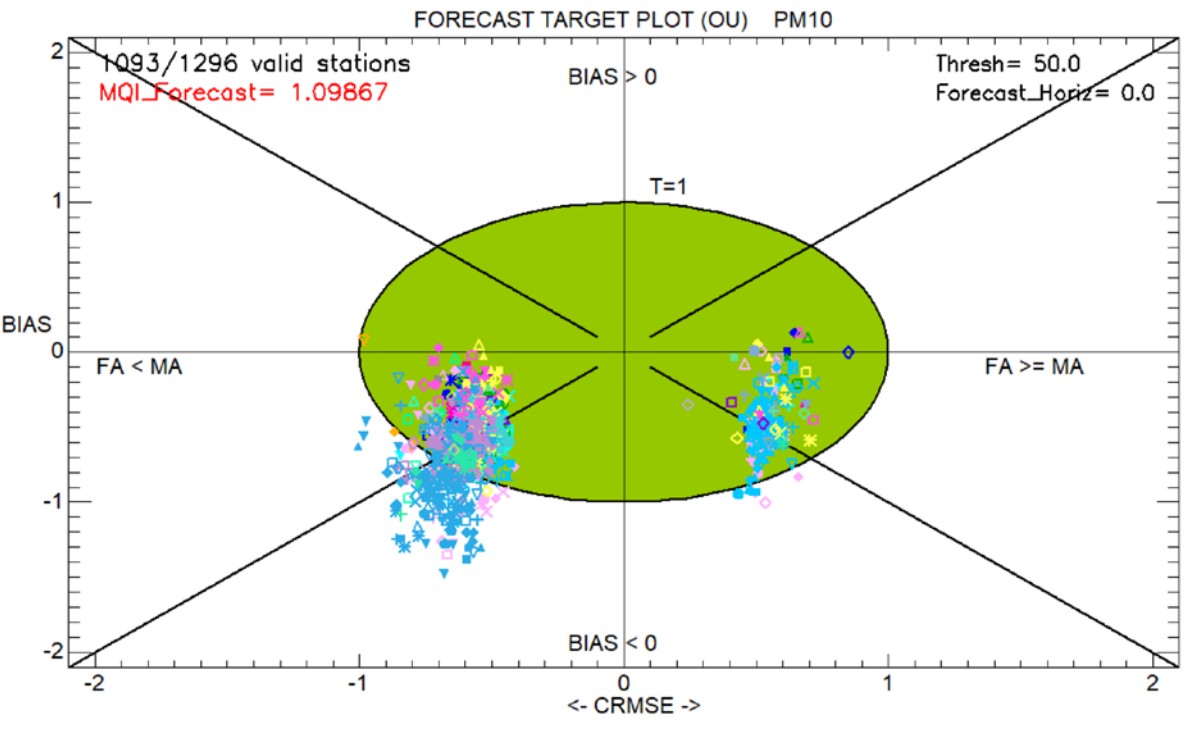

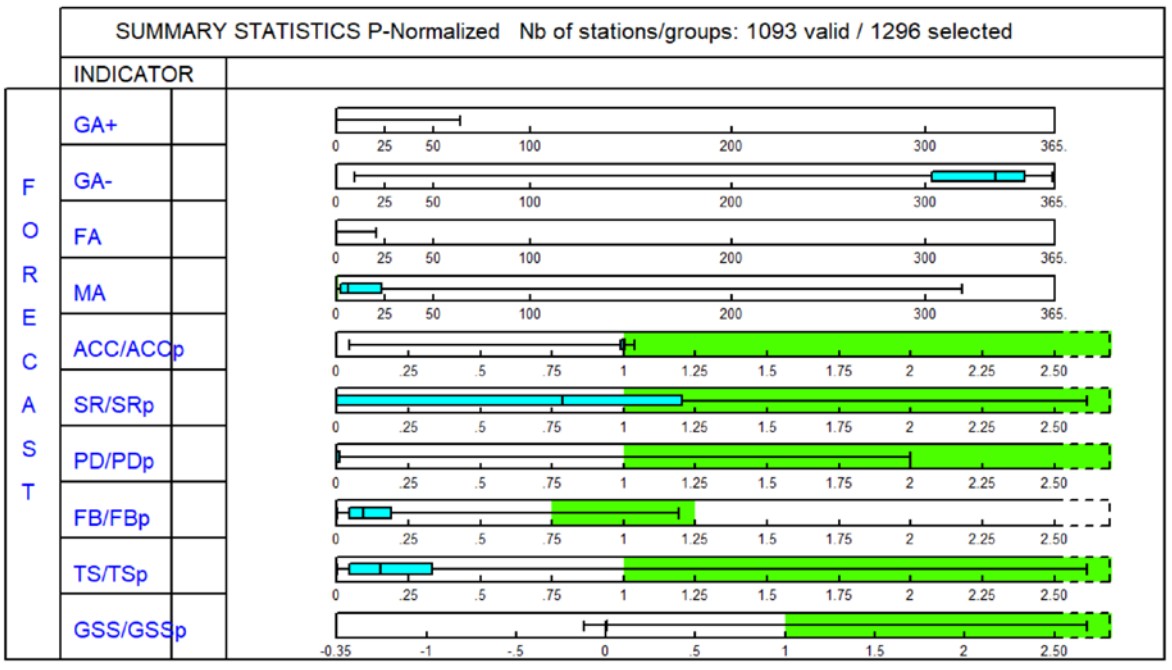

**Figure 1: FA1 validation outcomes for PM10. Forecast Target Plots (top) provide *MQI*ₓ values for each monitoring station, as the distance between the origin and a given point. Box plots in the Forecast Summary P-Normalized Reports (bottom) provide the statistical distribution (5th, 25th, 50th, 75th, 95th percentiles) of the categorical metrics.**

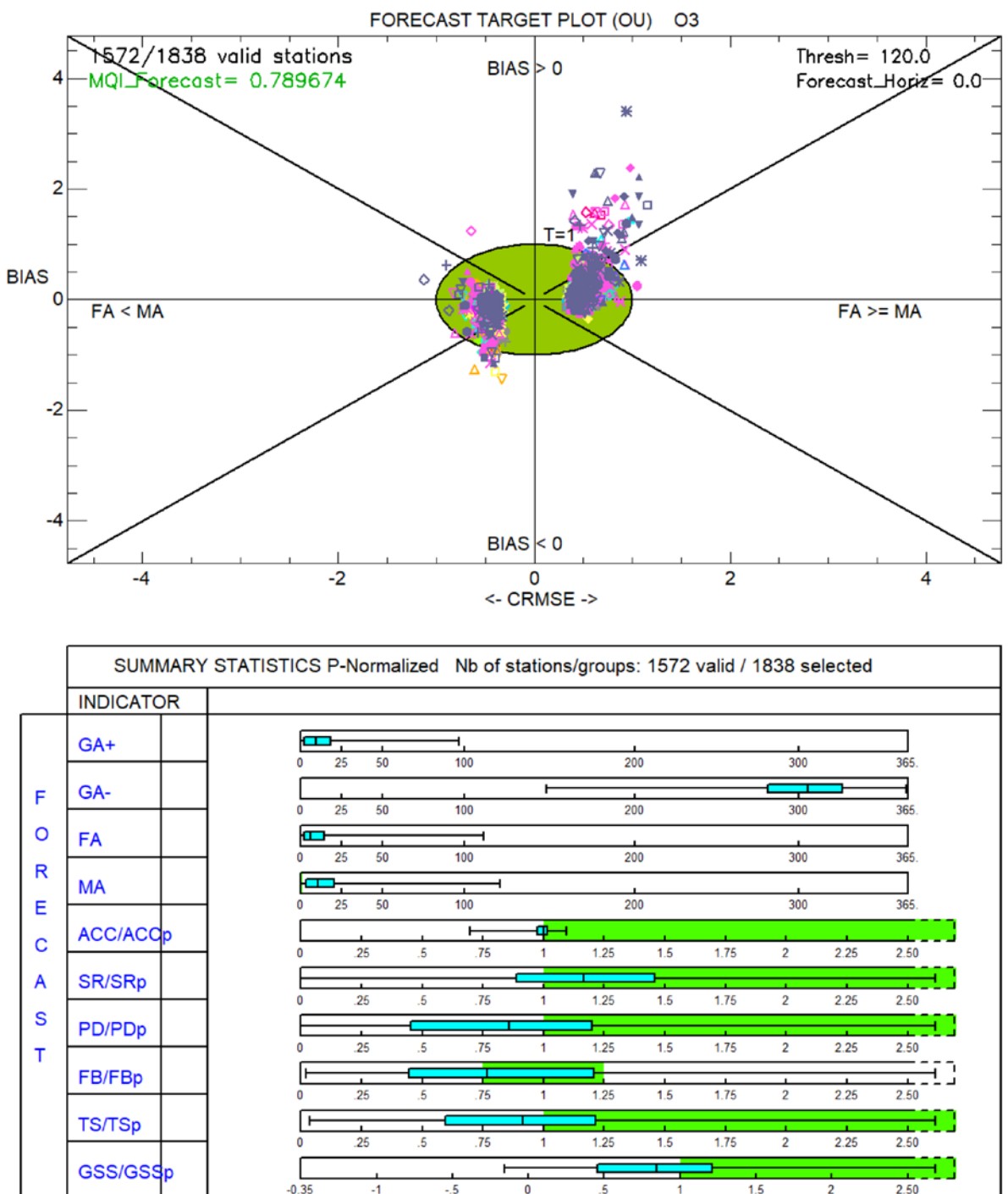

 **Figure 2: FA1 validation outcomes for O₃. Forecast Target Plots (top) provide** $MQI_f$ **values for each monitoring station, as the distance between the origin and a given point. Box plots in the Forecast Summary P-Normalized Reports (bottom) provide the statistical distribution (5th, 25th, 50th, 75th, 95th percentiles) of the categorical metrics.**

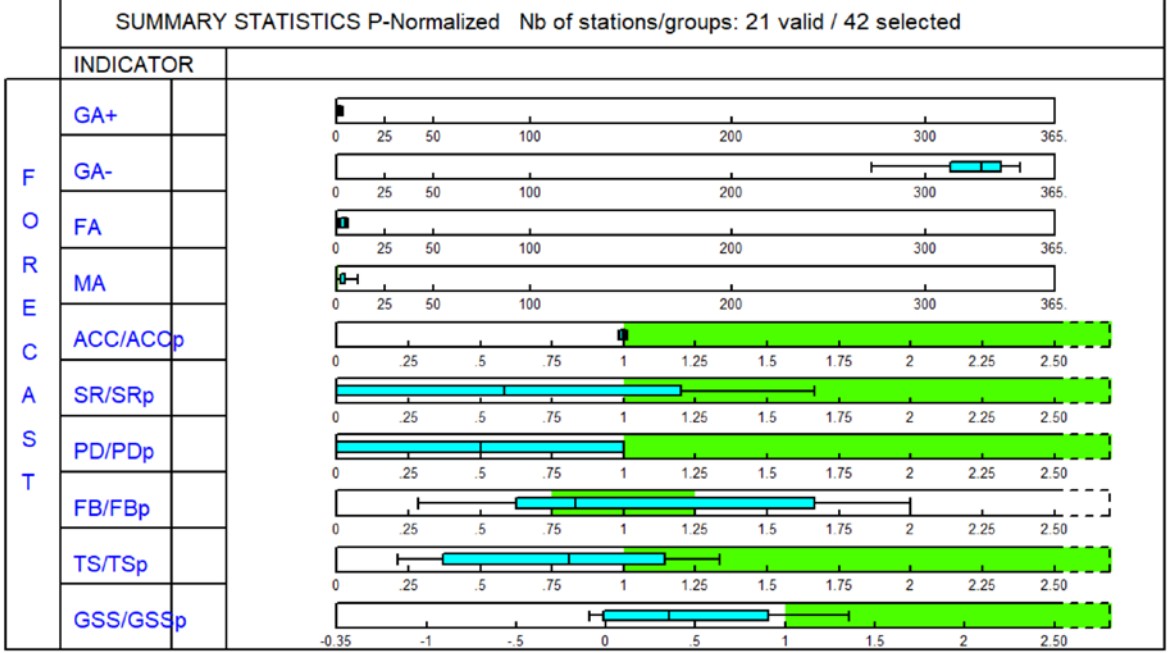

**Figure 3: FA2 validation outcomes for PM10.** Forecast Target Plots (top) provide *MQI*$_f$ values for each monitoring station, as the distance between the origin and a given point. Box plots in the Forecast Summary P-Normalized Reports (bottom) provide the statistical distribution (5$^{th}$, 25$^{th}$, 50$^{th}$, 75$^{th}$, 95$^{th}$ percentiles) of the categorical metrics.

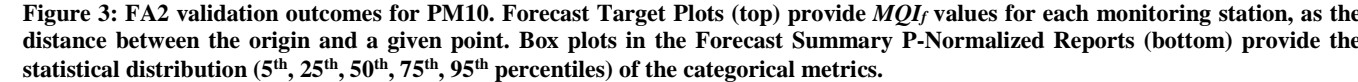

**Figure 4: FA2 validation outcomes for O₃. Forecast Target Plots (top) provide *MQI$_f$* values for each monitoring station, as the distance between the origin and a given point. Box plots in the Forecast Summary P-Normalized Reports (bottom) provide the statistical distribution (5ᵗʰ, 25ᵗʰ, 50ᵗʰ, 75ᵗʰ, 95ᵗʰ percentiles) of the categorical metrics.**

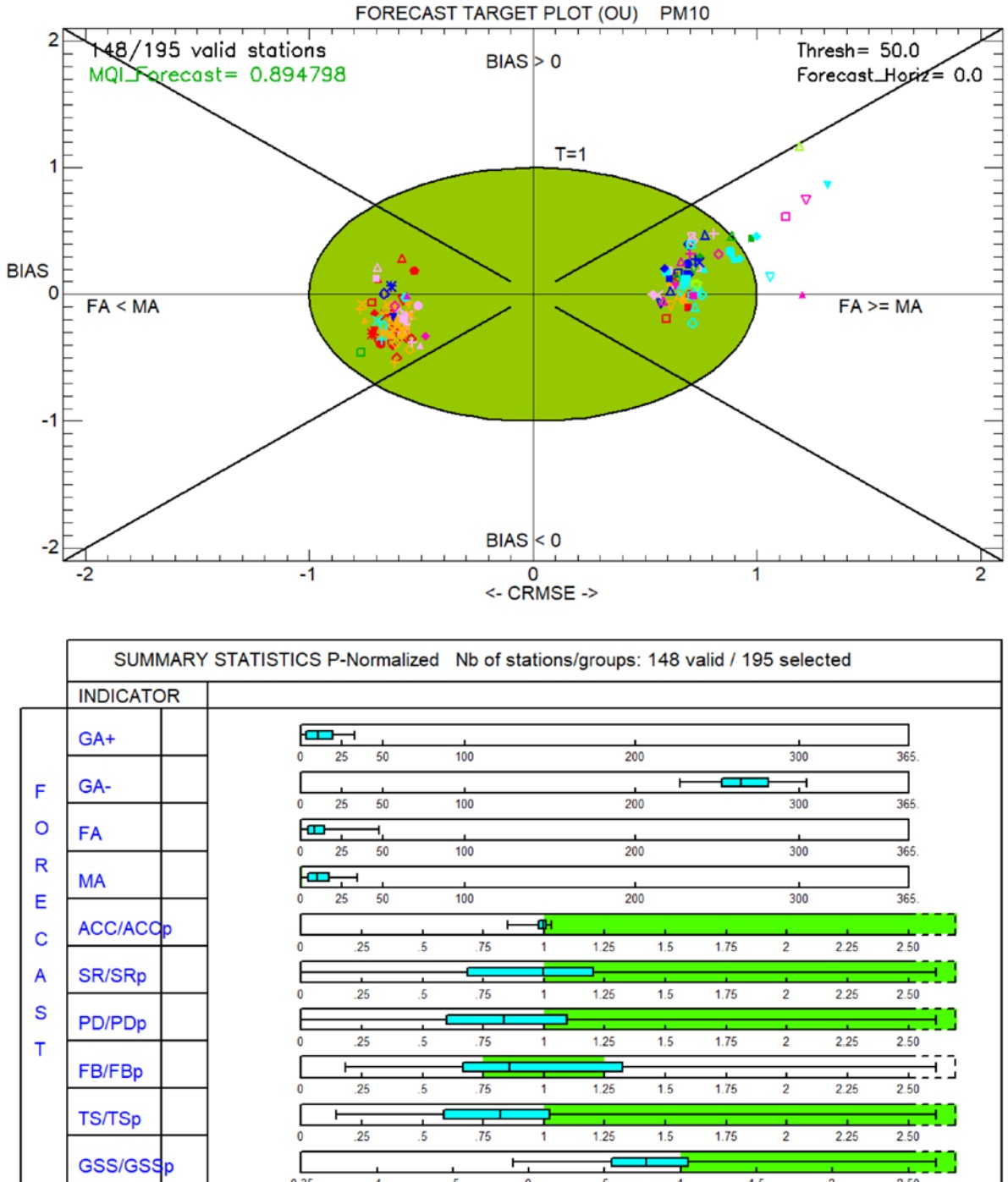

**Figure 5: FA4 validation outcomes for PM10.** Forecast Target Plots (top) provide *MQI_f* values for each monitoring station, as the distance between the origin and a given point. Box plots in the Forecast Summary P-Normalized Reports (bottom) provide the statistical distribution (5th, 25th, 50th, 75th, 95th percentiles) of the categorical metrics.

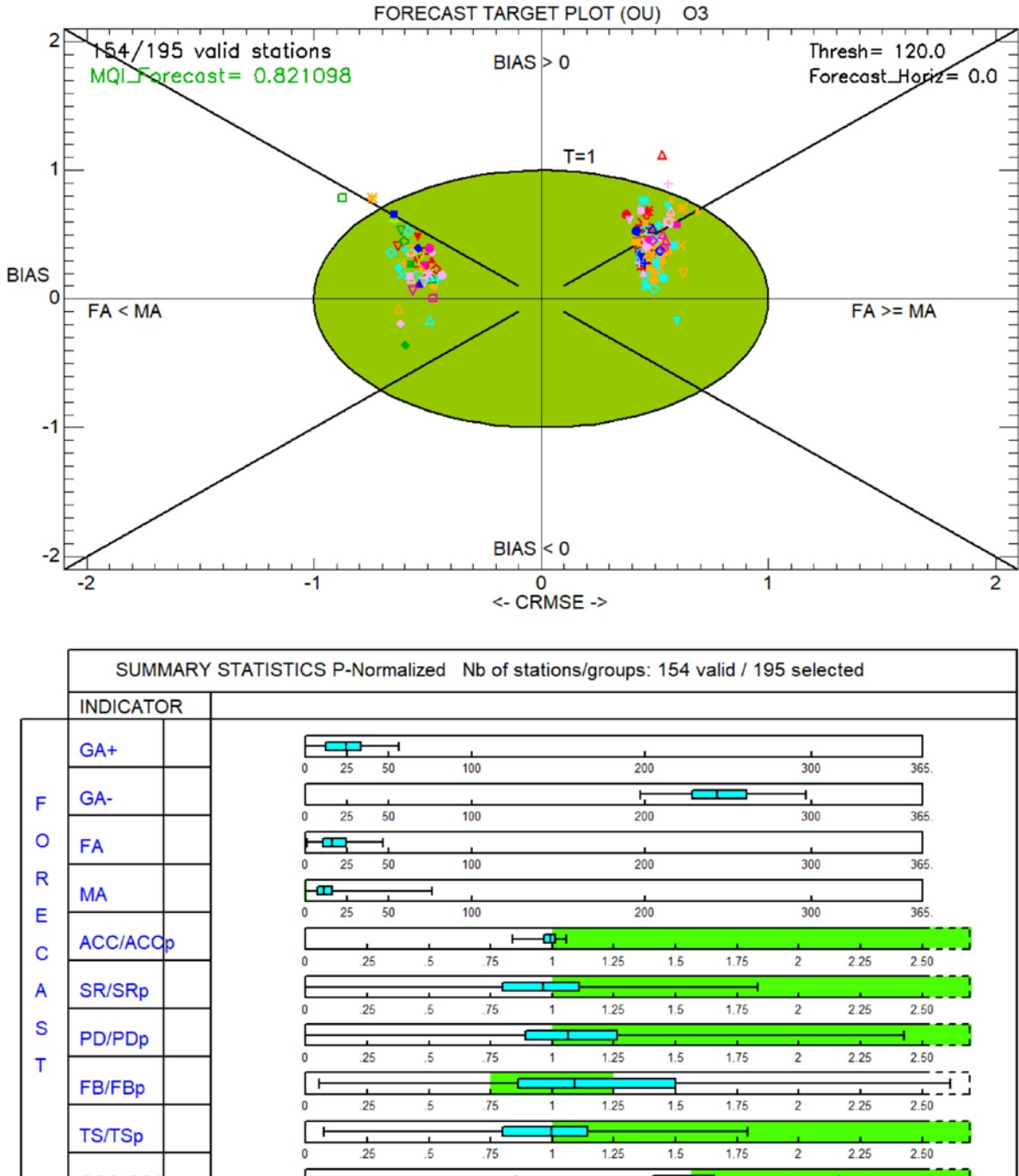

**Figure 6: FA4 validation outcomes for O₃. Forecast Target Plots (top) provide $MQI_f$ values for each monitoring station, as the distance between the origin and a given point. Box plots in the Forecast Summary P-Normalized Reports (bottom) provide the statistical distribution ($5^{th}$, $25^{th}$, $50^{th}$, $75^{th}$, $95^{th}$ percentiles) of the categorical metrics.**

Forecast Target Plots outcomes indicate a very good level of quality of all forecast applications in simulating O₃. The $90^{th}$ percentile of the $MQI_f$ values is lower than 1 for all three forecast applications, indicating that model performs better than persistence in simulating O₃ at more than 90% of the available stations. FA2 and FA4 fulfil $MQI_f$ requirements also in simulating PM10, instead there is room for improvement for the European scale simulation FA1 ($90^{th}$ percentile of the $MQI_f$ values is slightly higher than 1). Further investigations show that most of the issues emerge in a limited part of the modelling domain (Turkey), where very high, and sometimes unlikely, PM10 values are measured at several monitoring sites for most of the year. Removing Turkish monitoring stations from the validation data set, $MQO_f$ turns out to be fulfilled (Fig. D1, in Appendix D). It is worth noting that the $MQO_f$ outcomes are consistent with the standard assessment evaluation (Appendix C). Table C1 shows that the standard $MQO$ is fulfilled for all O₃ forecast applications. For PM10, the $MQI$ is higher than 1 but only for the FA1 simulation.

Concerning the capability in predicting exceedances, model performances improve moving from FA1 to FA4 applications (i.e. as spatial resolution increases) and skills are generally better in simulating O₃ than PM10. Concerning the comparison of the performances according to the different metrics, all forecast applications turn out to be better in avoiding false alarms than in reproducing all of them, since Success Ratio ($SR$) scores are generally better than Probability of Detection ($PD$) ones, especially for PM10.

In general, even if forecast applications are generally better than the persistence model according to the main outcome $MQO_f$ (top plots of Figs.1-6), it becomes harder for them to beat the persistence model in predicting exceedances (bottom plots of Figs.1-6). Apart from few cases (namely the regional FA4 application), the median values of the statistical distribution of the outcomes are not in the green area, indicating that model performs worse than persistence at more than 50% of the available stations.

## 4.2 MPI Plot supporting the interpretation of $MQO_f$ outcomes

When evaluating a forecasting application, it is of interest to assess the evolution of skills metrics with the forecast horizon. Indeed, a good forecasting application should not incur a substantial degradation of its performances along with forecast time.

FA3, carried out over Ireland by means of the OPAQ statistical system, was evaluated for each of the forecasted days, which included the current day (day0), tomorrow (day1) and the day after tomorrow (day2).

In the following it is reported how performances in simulating PM10 vary along with the forecast days. More in detail outcomes for day0 and day2 are shown in Fig. 7 and Fig. 8, respectively. On the top of each Figure, the Forecast Target Plots (described in the previous section) are reported. On the bottom, the Forecast MPI Plots are added, describing the fulfilment of both the criteria defined in Sect. 2.1 (i.e. $MPI$ less than or equal to 1). Indeed, here the forecast performances ($MFE_f$) are

compared to the persistence model performances ($MFE_p$) along Y axis ($MPI_1$) and to the Mean Fractional Uncertainty ($MF_U$) along the X axis ($MPI_2$). The green area identifies the area of fulfilment of both proposed criteria. The orange areas indicate where only one of them is fulfilled.

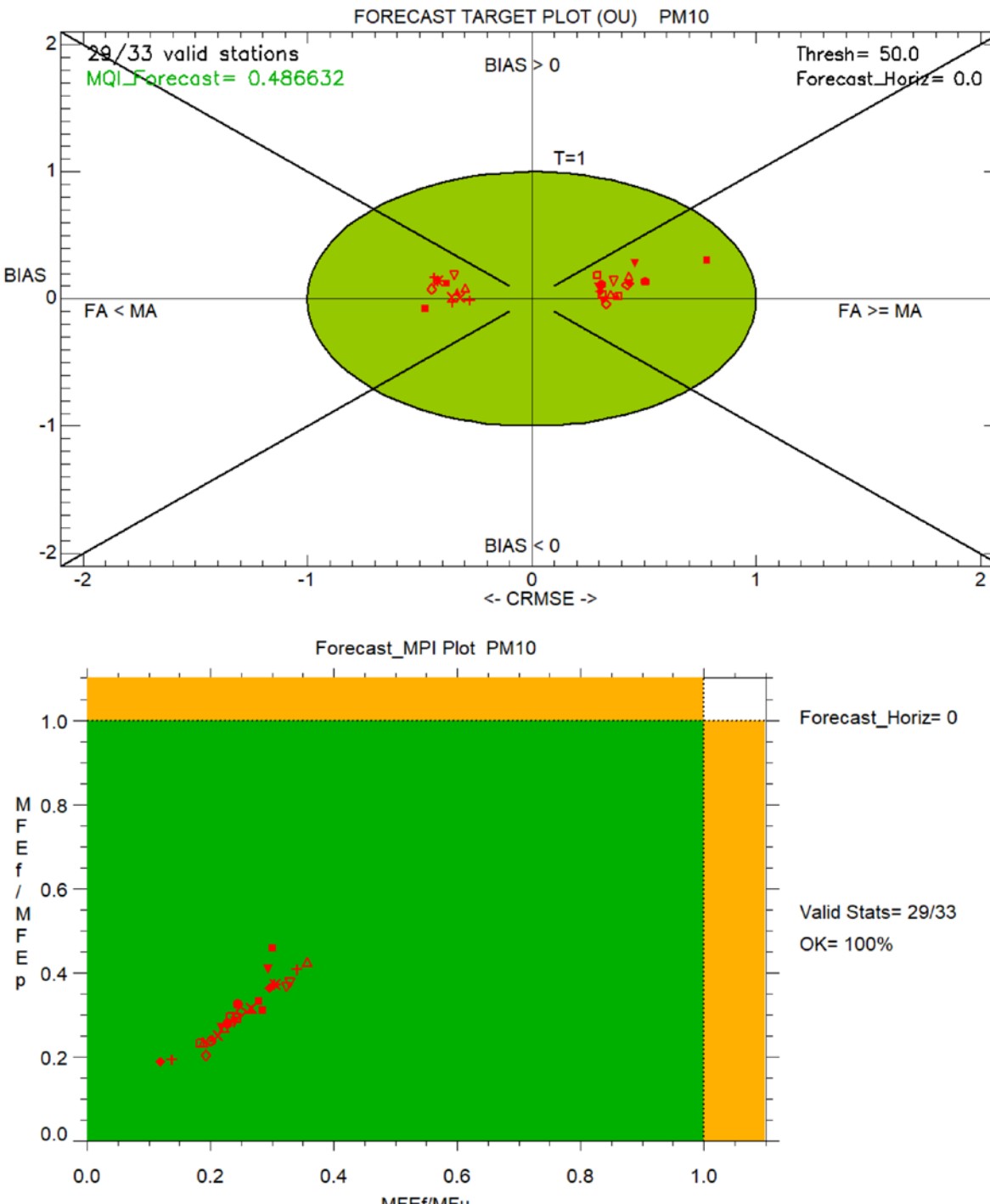

Figure 7: FA3 validation outcomes for day0. Forecast Target Plots (top) provide $MQI_f$ values for each monitoring station, as the distance between the origin and a given point. Forecast MPI Plots (bottom) provide for each monitoring station $MPI_1$ along Y axis and $MPI_2$ along the X axis.

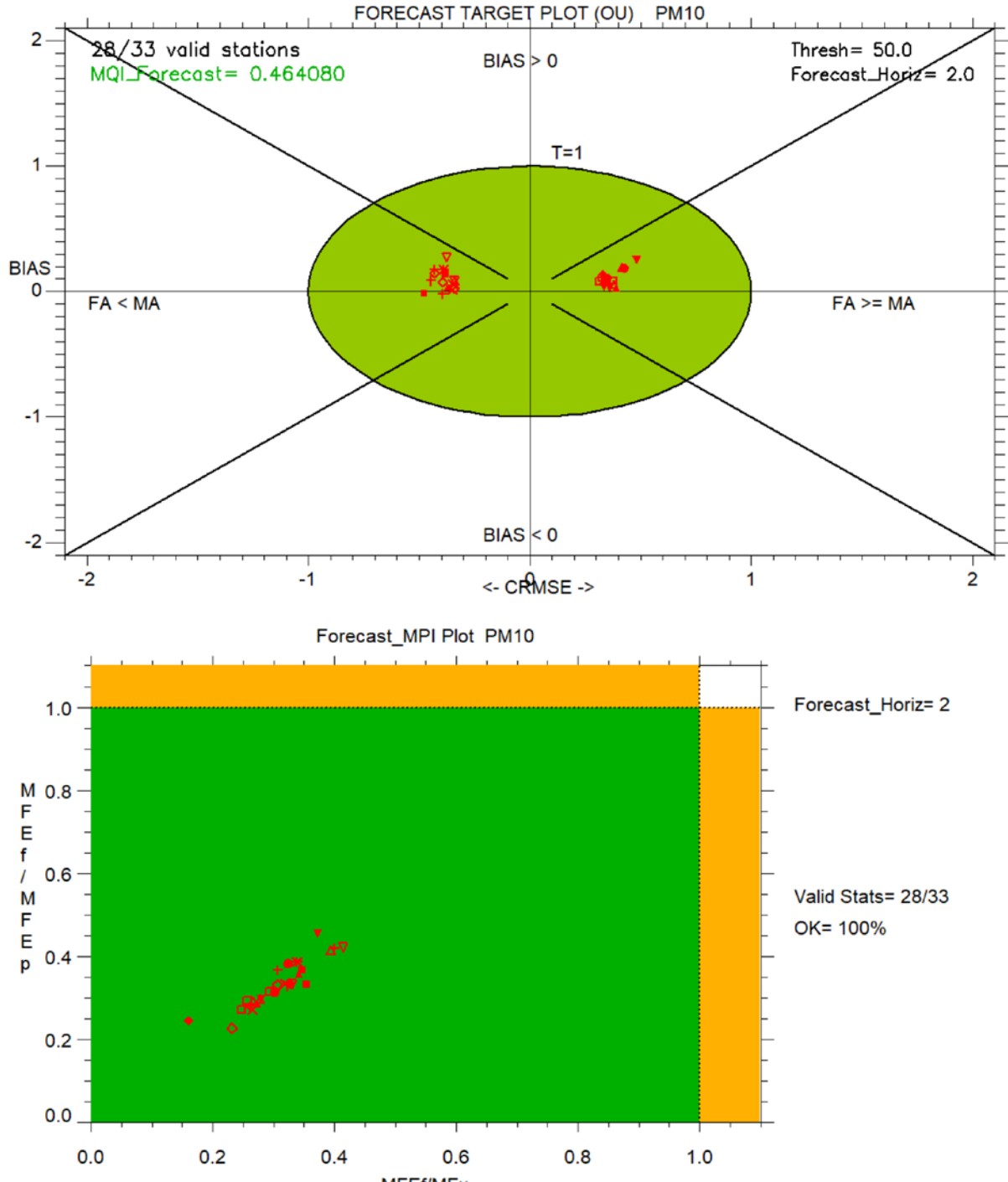

**Figure 8: FA3 validation outcomes for day2. Forecast Target Plots (top) provide *MQI$_f$* values for each monitoring station, as the distance between the origin and a given point. Forecast MPI Plots (bottom) provide for each monitoring station *MPI$_1$* along Y axis and *MPI$_2$* along the X axis.**

Outcomes in Figs. 7-8 indicate a very good level of quality of the forecast application, since Modelling Quality Objective is fulfilled (top), together with the two additional Performance Criteria (bottom). These outcomes are consistent with the standard $MQO$ skills provided in Table C1 of Appendix C, which points out very good performances of FA3 for PM10, namely the best performances among all forecast applications.

Concerning the evolution of skills metrics with forecast horizon, according to the Forecast Target Plot outcomes (top), modelling performances unexpectedly get better from day0 to day2, since the $MQI_f$ value associated to the 90th percentile worst station (reported in the upper left corner of the plots) turns out to get lower. According to the Forecast MPI Plots (bottom), performances remain almost constant with forecast horizon, indicative of a good behaviour of the modelling application. Moreover, Forecast MPI Plots help to clarify that the unrealistic improvement of model performances from day0 to day2, pointed out by the Forecast Target Plots, is due to persistence model performances degradation. Indeed, moving from day0 to day2, the forecast model performances get slightly better along Y axis, where they are normalized to persistence model skills, but they slightly deteriorate along X axis, where they are considered regardless of persistence aspects. In other words, model performances slightly deteriorate along with the forecast days but persistence model deteriorate more, so that performances ratios (i.e. both $MQI_f$ and $MPI_1$ values) get lower.

## 4.3 Assessment of modelling application capability in predicting Air Quality Indices

The current approach for the assessment of modelling application capability in predicting Air Quality Indices is based on a cumulative analysis for answering the following questions: "Are citizens correctly warned against high pollution episodes?" or in another words: "Does the model properly forecast AQI levels?"

Air Quality Indices are designed to provide information on local air quality. Moreover, within the proposed validation protocol, the capability of correctly predicting AQI is assessed at single monitoring stations. For these reasons, FA5 at the local scale is the most suitable for testing the proposed approach. Indeed, it was carried out at high spatial resolution and focuses on only two monitoring sites, located in two cities in Kosovo: Pristine (the capital) and Drenas.

Before analysing AQI results for PM2.5, it has to be mentioned that the FA5 standard $MQO$ is fulfilled for all available pollutants (Table C1 in Appendix C). Concerning additional features of the forecasting validation protocol, both the Forecast Target Plot and the Forecast MPI Plot show very good performances for both locations. The Forecast Summary P-Normalized Report indicates good model performance in Drenas and some room for improvements in Pristine location due to underestimation of PM2.5 episodes.

Fig. 9 provides the AQI diagram, based on EEA classification, for PM2.5 and for day0 forecast. For each station, the bar plot shows two paired bars: the number of predicted (left bar) and measured (right bar) concentration values that fall within a given air quality category. In Drenas, forecast values populate categories 2 ("Good"), 3 ("Medium"), and 4 ("Poor") to a greater extent than the measurements. On the contrary, in Pristine forecast values are more frequent than the measurements at the lowest AQI ("Very Good").

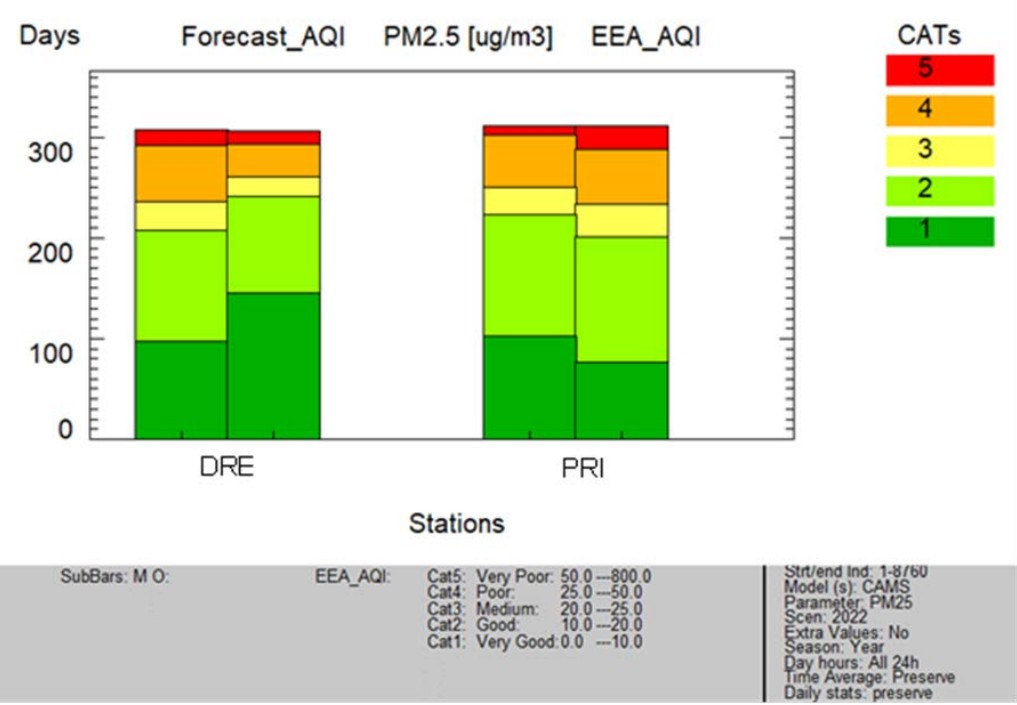

**Figure 9: FA5 validation outcomes for PM2.5 at Drenas and Pristina. AQI diagram provide for each monitoring station the number of predicted (left bar) and measured (right bar) concentration values that fall within each air quality category. The last two EEA AQI classes ("Very poor" and "Extremely poor") are merged into one**.

Overall, Fig. 9 points out that FA5 generally overestimates PM2.5 concentration levels in Drenas and underestimates them in Pristine. Anyway, AQI forecast bar plots provide information about the total number of occurrences in each AQI class but there is no information about the correct timing of the forecasted AQI level.

So, there is room for future improvement and other additional outputs could be included within the protocol. In particular, multi-category contingency tables can be created for each station and multi-categorical skill scores can be computed, according to literature (e.g. EPA, 2003). Outcomes can be plotted for single stations or describing, for each AQI class, skill scores statistical distribution among the stations.

For example, in Fig. 10 an in-depth insight of AQI assessment is proposed for Drenas (top) and Pristina (bottom). Two additional multi-categorical metrics are proposed. Both of them are computed for each AQI level and are based on the comparison between forecast and measurement values considering also the correct timing of the predicted AQI level. *AQI comparability* (left plots in Fig. 10) represents, for each of the five AQI classes, the percentage of the correct forecast events in this class with respect to the total events based on measurements. Since *AQI comparability* values are percentages, they range from 0 to 100, being 100 the optimal value. *TS_AQI* (right plots in Fig. 10) is computed according to the same

definition of *TS* in Table 1. Indeed, here multiple thresholds (i.e. class limits) are taken into account and so multiple outcomes, one for each AQI class, are provided. *TS_AQI* values range from 0 to 1, being 1 the optimal value.

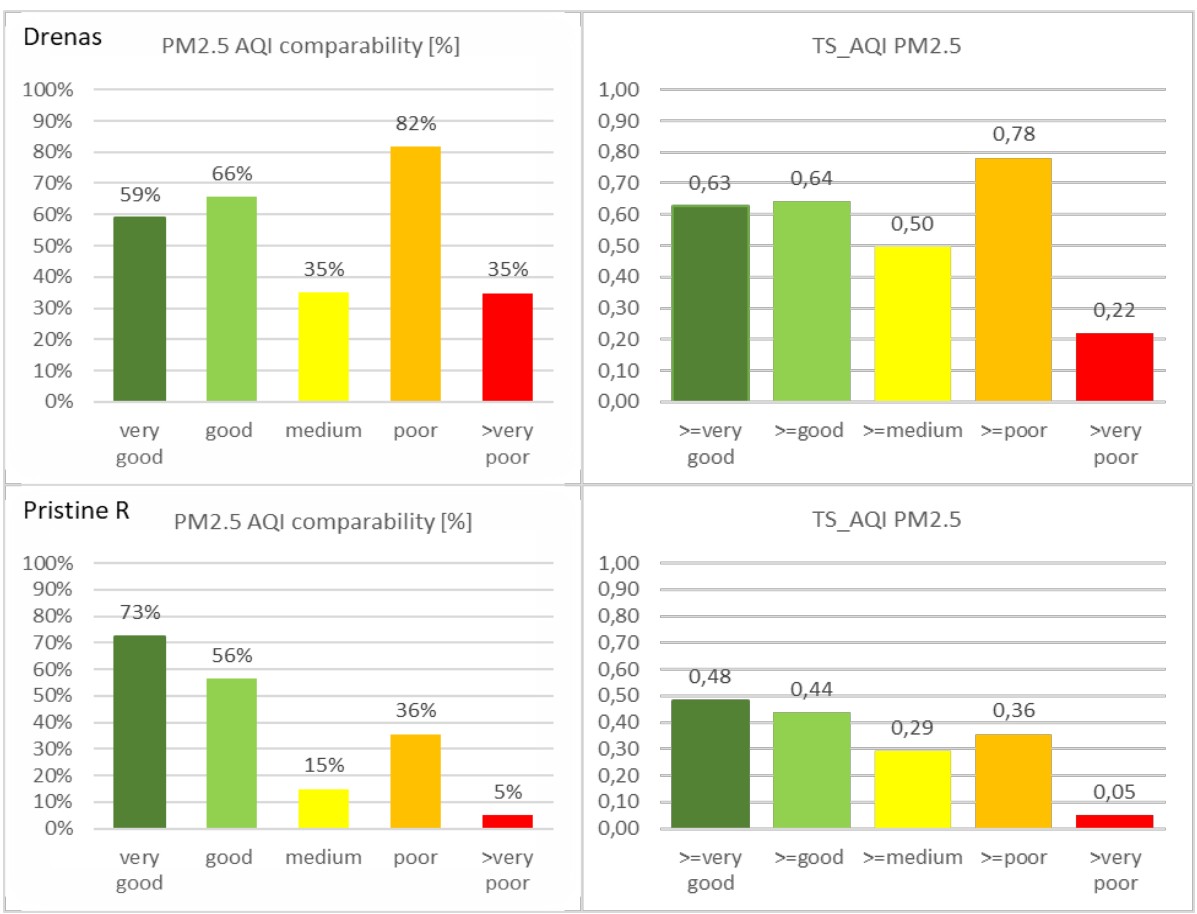

**Figure 10: Multi-categorical metrics outcomes for Drenas (top) and Pristina (bottom). AQI comparability plots (left) provide for**
**each AQI class the percentage of the correct forecast events with respect to the total events based on measurements. TS_AQI plots**
**(right) provide for each AQI class *TS_AQI* values.**

*AQI Comparability* and *TS_AQI* in Fig. 10 provide additional information with respect to AQI diagram. For example, in the case of Drenas, it turns out that, according to both the metrics, the best agreement between forecast and measurements in predicting the correct timing of the occurrences are found for "Poor" AQI class. It is also worth noting that, even if
according to cumulative analysis (Fig. 9) forecast and measurements present a similar number of occurrences in both the "Medium" and the "Very Poor" classes, according to *AQI Comparability*, these classes are characterized by the worst performances. *TS_AQI* gives additional information about the model performances, which is especially noticeable for the "Medium" and "Very poor" classes, as it defines levels differently ("Medium" class means "Medium" class and all higher classes – "Poor" and "Very poor"). In this case the "Medium" class is characterized by better performances than the "Very

Poor" class. In the case of Pristine location, the best performances, according to both the metrics, are achieved for low concentrations ("Very Good" and "Good" classes) and the worst ones for "Very Poor" and "Medium" AQI levels. It is also worth noting that the best agreement is found for "Good" class, according to the cumulative comparison (Fig. 59), but it is better for "Very Good" class if the timing of the occurrences is taken into account (Fig. 10).

## 4.4 Discussion

Several lessons were learnt from the results presented here. The main proposed criterion ($MQO_f$) turned out to be useful for evaluating the strengths and the shortcomings of a forecasting application, focusing on features which could not be addressed with  the assessment evaluation approach.

Side outcomes, included within the protocol, can help in deepening the analysis. For example, $MPIs$ analysis based on $MFE$ helps in interpreting the outcomes, since $MPI_2$ is formulated regardless of persistence aspects, providing details on the model

performances.

Consistently with the FAIRMODE approach, the measurement uncertainty is considered within the $MQO_f$ formulation. While values are currently based on maximum uncertainties (95$^{th}$ percentile), these could be modified in the future to obtain a consensus level of stringency for the $MQO_f$, i.e. a level reachable for best applications while stringent enough to preserve sufficient quality. In Appendix E the outcomes of a sensitivity analysis are provided, in which we investigate the impact of

the value chosen as representative measurement uncertainty.

Concerning the capability in predicting the exceedances, it turned out that, regardless of the spatial scale and the pollutants, even if a forecast application is better than the persistence model according to the general evaluation criterion ($MQO_f$), it can be worse in correctly providing categorical answers. Indeed, the difficulty in beating the persistence model skills is not infrequent in weather forecasting applications (Mittermaier, 2008). Moreover, it is worth noting that, differently from $MQO_f$

analysis, the evaluation of the model capability in predicting the exceedances, being based on the definition of fixed thresholds, does not take into account the measurement uncertainty. For these reasons, a "fitness for purpose" criterion concerning exceedances metrics (e.g., which percentiles of a categorical indicator should be in the green area in order to define its skill "good enough"? and following on from that, how many indicators should be "good enough" in order to define the forecast application "fit for purpose"?) is not definitively set within the proposed protocol. Indeed, some more discussion

based on further tests on forecasting applications are needed.

The greatest room for improvement turns out concerning the evaluation of the capability of the forecasting application in predicting AQI levels. The current approach is based on a cumulative analysis and no information is provided about the correct timing of the forecasted AQI levels. To account for this, some preliminary tests were carried out based on two additional multi-categorical metrics, which sound interesting in complementing the current approach. The main weakness of

the proposed approach is the large number of different values to be provided, so making this type of outcome usable only for single monitoring stations. Moreover, the question of which level of performances in AQI predicting is "good enough" is currently an open issue and benchmarking of several forecasting applications is needed to establish some quality criteria.

**5 Conclusions**

A standardized validation protocol for air quality forecast applications was proposed, following FAIRMODE community
discussions on how to address specific issues typical of forecasting applications.

The proposal of a common benchmarking framework for model developers and users supporting policymaking under the European Air Quality Directives is a major achievement.

The proposed validation protocol enables an objective assessment of the "fitness for purpose" of a forecasting application, since it relies on the usage of a reference forecast as a benchmark (i.e. the persistence model), includes the measurement
uncertainty, and bases the evaluation on the fulfilment of specific performance criteria, defining an acceptable quality level of the given model application. On top of a pass/fail test to ensure fitness-for-purpose (intended as a necessary but not sufficient condition), a series of indicators is proposed to further analyse the strengths and weaknesses of the forecast application.

Moreover, relying on a common standardized validation protocol, the comparison of performances of different forecast
applications, within a common benchmarking framework, is made available.

The application of the methodology to validate several forecasting simulations across Europe, using different modelling systems and covering various geographical contexts and spatial scales, suggested some general considerations about its usefulness.

The main "fitness for purpose" criterion, describing the global performances of the model application with respect to
persistence skills, proves to be useful for a comprehensive evaluation of the strengths and the shortcomings of a forecasting application. Generally, the forecast Modelling Quality Objective turns out to be achievable for most of the examined validation exercises. When the criterion was not addressed, side analyses and outcomes, included within the protocol, helped in deepening the analysis and in identifying the most critical issues of the forecasting application.

On the other hand, it turned out that, regardless of the spatial scale and the pollutants, it can be hard for a forecast application
beating persistence model skills in correctly providing categorical answers, namely on exceedances of concentration thresholds. Therefore, further tests and analyses are needed in order to provide some criteria for defining the "fitness for purpose" of a forecasting application in predicting exceedances.

The last model capability assessed within the proposed validation protocol concerns the correct prediction of Air Quality Indices, designed to provide citizens with effective and simple information about air quality and its impact on their health.
The current approach is based on a cumulative analysis of relative distributions of observed and forecasted AQIs. As no information is provided about the correct timing of the forecasted AQI levels, further developments are foreseen based on multi-category contingency tables and multi-categorical skill scores.

Actually, discussion on the proposed approach will go on within the FAIRMODE community and upgrades and improvements of the current validation protocol will be fostered by its usage. In particular, it will be of interest to collect
feedback from in-depth diagnostic analyses focusing on the validation of specific forecast applications, using both the

proposed criteria and the threshold-based categorical metrics to gain further insights. Anyway, from its preliminary applications across Europe, the methodology turns out to be sufficiently robust  for testing and application, especially targeting air quality forecasting services supporting policymaking in European Member States.

## Appendix A

Measurement uncertainty $U(O_i)$ as a function of the concentration values $O_i$ can be expressed as follow:

$$U(O_i) = U_r(RV)\sqrt{(1-\alpha^2)O_i^2 + \alpha^2 RV^2} \qquad (A1)$$

An in-depth description of the rationale and the formulation of the measurement uncertainty estimation is provided in Thunis et al. (2013) and Pernigotti et al. (2013) for $O_3$, and PM and $NO_2$, respectively. More in details, the formulation of the measurement uncertainty as a function of the measured concentration is based on two coefficients: $U_r(RV)$, i.e. the relative

uncertainty around a reference value $RV$ and $\alpha$, i.e. the fraction of uncertainty non proportional to the concentration value. It is important to note that we use as representative for the measurement uncertainty the 95th percentile highest value among all uncertainty values. For PM10 and PM2.5 the results of a JRC instrument inter-comparison (Pernigotti et al., 2013) have been used whereas a set of EU AIRBASE stations available for a series of meteorological years has been used for $NO_2$ and analytical relationships have been used for $O_3$. These 95th percentile uncertainties only include the instrumental error.

Parameters $U_r(RV)$, $RV$, and $\alpha$ for $U(O_i)$ calculation for $NO_2$, $O_3$ and PM are provided in Table A1.

**Table A1** Parameters for the calculation of measurement uncertainty.

|  | $U_r(RV)$ | $RV$ | $\alpha$ |
|---|---|---|---|
| **NO₂** | 0.24 | 200 µg/m3 | 0.20 |
| **O₃** | 0.18 | 120 µg/m3 | 0.79 |
| **PM10** | 0.28 | 50 µg/m3 | 0.25 |
| **PM2.5** | 0.36 | 25 µg/m3 | 0.50 |

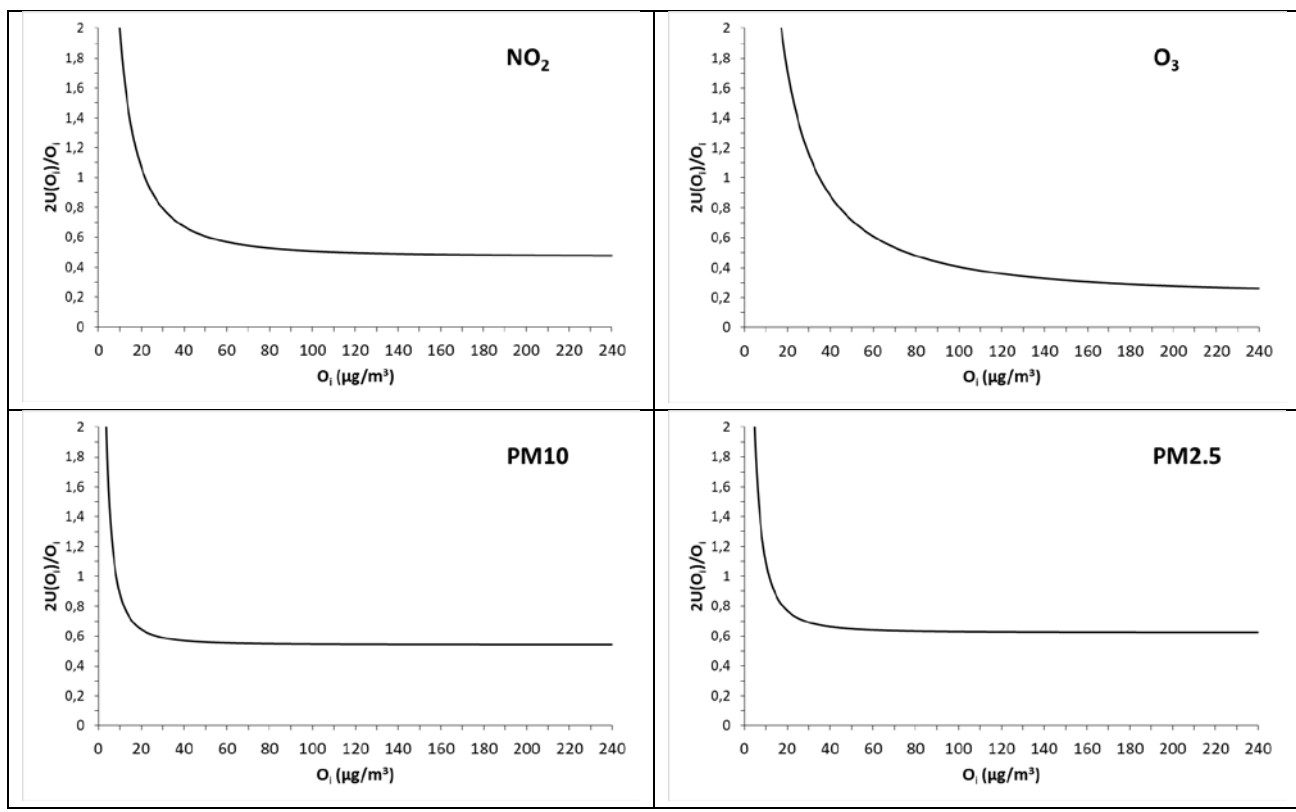


Figure A1: Double relative measurement uncertainties as a function of concentration values for NO$_2$ (top, left), O$_3$ (top, right), PM10 (bottom, left) and PM2.5 (bottom, right).

**Appendix B**

Table B1 Contingency Table.

| Forecast Events | Yes | FA | GA+ |
|---|---|---|---|
| | No | GA- | MA |
| | | No | Yes |
| CONTINGENCY TABLE | | Observed Events | |

 **Appendix C**

The standard Modelling Quality Objective (*MQO*), valid for assessment, is defined by the comparison of model-observation differences (namely, the root mean square error, *RMSE*) with a quantity proportional to the measurement uncertainty.

$$MQI = \frac{RMSE}{\beta \sqrt{\frac{\sum_{i=1}^{N}\left(U(O_i)\right)^2}{N}}}$$ (C1)

β is set to 2, thus allowing the deviation between modelled and observed concentrations to be twice the measurement uncertainty. Measurement uncertainty $U(O_i)$ as a function of the concentration values $O_i$ is defined in Appendix A.

The *MQO* is fulfilled when *MQI* is less than or equal to 1.

Standard assessment *MQO* outcomes (i.e., *MQI* value associated to the 90[th] percentile worst station) for all available pollutants are summarized in Table C1 for all forecast applications.

**Table C1** Standard assessment *MQI* values (associated to the 90[th] percentile worst station) for all forecast applications.

|      | NO$_2$ | O$_3$ | PM10  | PM2.5 |
|------|--------|-------|-------|-------|
| FA1  | 0.865  | 0.619 | 1.267 | 0.776 |
| FA2  | 0.831  | 0.698 | 0.941 | 0.700 |
| FA3  |        |       | 0.479 |       |
| FA4  | 1.009  | 0.696 | 0.943 | 1.009 |
| FA5  | 0.685  | 0.570 |       | 0.528 |

**Appendix D**

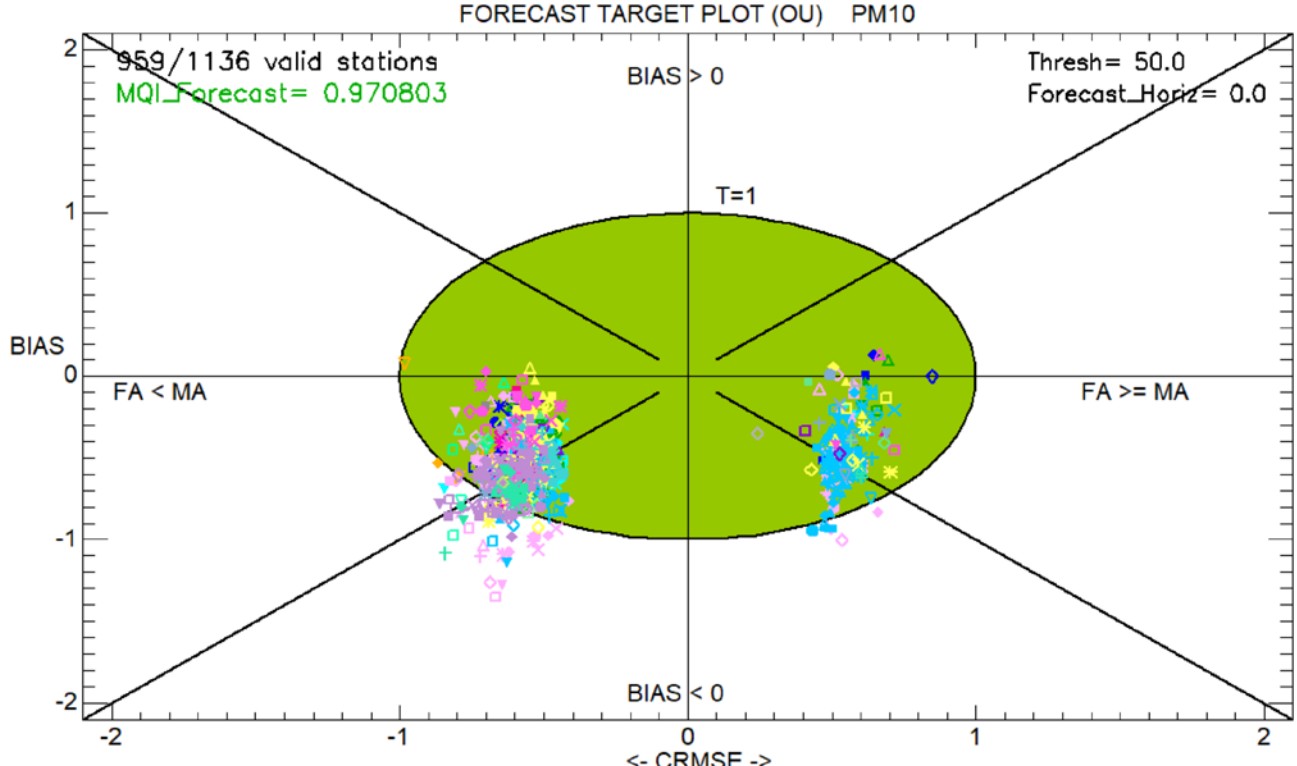

**Figure D1: FA1 Forecast Target Plot for PM10, removing Turkish monitoring stations from the validation data set.**

**Appendix E**

The effect on $MQI_f$ outcomes of lowering measurement uncertainty estimates is investigated here. More in detail, the values of $U_r(RV)$ parameters in Table A1 (i.e. the estimates of the relative uncertainty around the reference value, defining the asymptotic behaviour of the functions of Figure A1) were reduced by 25% and 50% for all the pollutants and the $MQI_f$ were

recalculated for the different forecast applications. Figure E1 shows the results for all available data: FA1, FA2, FA4 outcomes for the current forecast day (all pollutants available), and FA3 outcomes along a three-days forecast horizon (only PM10 available). Different colours refer to results based on different $U_r(RV)$ values: 1 $U_r(RV)$ indicates the original values in Table A1, 0.75 $U_r(RV)$ and 0.50 $U_r(RV)$ refer to 25% and 50% reductions, respectively. Indeed, the 50% reduction decreases the $U_r(RV)$ values to 0.12 ($NO_2$), 0.09 ($O_3$), 0.14 (PM10), 0.18 (PM2.5), i.e. well below the data quality objective

values set by the current European legislation (European Union, 2008), namely 15%, 15% and 25% for $NO_2$, $O_3$, and particulate matter.

The results of the sensitivity analysis are provided by means of the violin plots (Hintze and Nelson, 1998), showing the distributions of the $MQI_f$ values computed for each monitoring station. In other words, each violin refers to all the data

provided within the corresponding Forecast Target Plot, giving in a single plot an overall view of all the outcomes available.

Three lines were added to the display of each violin, indicating the 10th, the 50th and 90th percentiles of the distributions.

Results show that both $MQI_f$ values and the shape of their distribution depend on both the forecast application and the pollutant. Within this context, changing $U_r(RV)$ values induces a very slight effect on the shape of the $MQI_f$ values distribution, apart from the case of PM2.5 for FA2, where a small amount of data is available (11 monitoring stations). On the contrary, as expected, changing $U_r(RV)$ values turns out in variations of $MQI_f$ values, which get higher as $U_r(RV)$ gets

lower, to a different extent depending on the forecast application and the pollutant. Generally, variations tend to be lower if data availability is higher. Concerning the main $MQO_f$ criterion fulfilment (i.e. the 90th percentile of the $MQI_f$ values is lower than 1), being based on a categorical answer (yes/no), it changes or not mainly depending on the performances of the reference analysis (1 $U_r(RV)$). The same answer is maintained both in case of very good performances ($MQI_f$ 90th percentile value largely lower than 1) and in case the criterion is not fulfilled even in the reference analysis ($MQI_f$ 90th percentile value

already higher than 1). When $MQI_f$ 90th percentile value is lower but quite close to 1, $MQO_f$ criterion fulfilment is of course more sensitive to measurement uncertainty estimate. Indeed, this is expected and it is a typical shortcoming of the usage of criteria based on categorical answers.

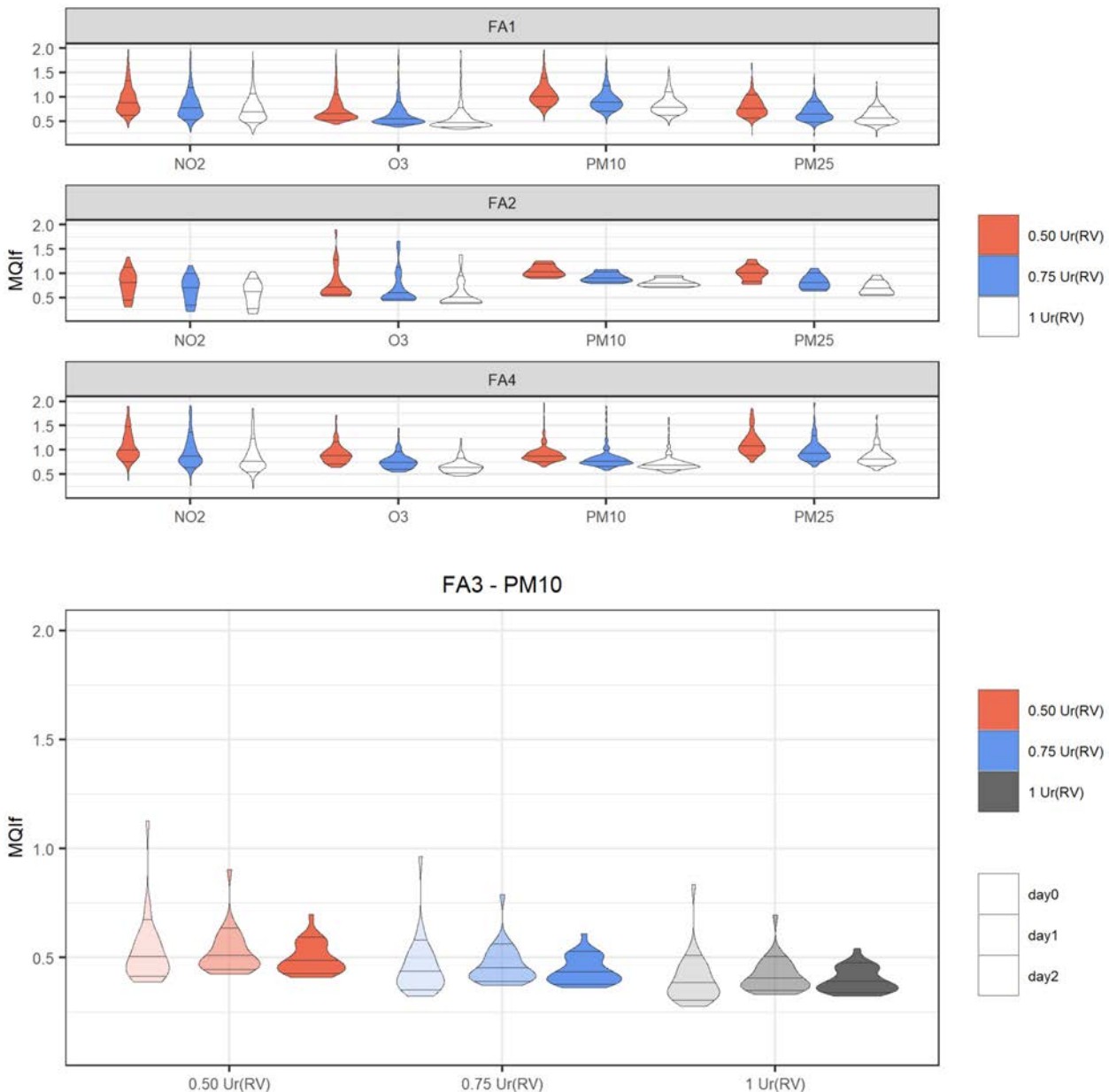


**Figure E1: Effect on the distribution of *MQI<sub>f</sub>* values of lowering *U<sub>r</sub>(RV)*. Top: FA1, FA2, FA4 outcomes for the current forecast day (all the pollutants available). Bottom: FA3 outcomes along a three-days forecast horizon (only PM10 available).**

*Code and data availability.* The DELTA Tool software and all datasets generated and analysed during the current study are
480  available on Zenodo at https://doi.org/10.5281/zenodo.7949868.

*Author contributions.* KC, PD, SJ, AM, AP, PT and LV contributed to the study conception and design. PT conceived part of the methodology and KC developed the software. Material preparation, data collection and analysis were performed by MA, RA, AB, AD, CG, GG, AM, TP, MS, LV and SV. The first draft of the manuscript was written by AM, AP and LV and all authors commented on previous versions of the manuscript. All authors read and approved the final manuscript.

*Competing interests.* The authors declare that they have no conflict of interest.

*Acknowledgements.* Thanks are due to FCT/MCTES for the contract granted to Carla Gama (2021.00732.CEECIND) and for the financial support to the CESAM Associated Laboratory (UIDB/50017/2020 + UIDP/50017/2020).

The analysis for the Irish air quality forecasts has been supported by the LIFE Emerald project "Emissions Modelling and Forecasting of Air in Ireland" which is co-funded by the European Union, under grant agreement No. LIFE19 GIE/IE/001101. VITO thanks the Irish EPA for sharing all observation data and continuous valuable feedback on the model developments.

NINFA simulation over Po Valley and Slovenia was developed under the project LIFE-IP PREPAIR (Po Regions Engaged to Policies of AIR), which was co-funded by the European Union LIFE Program, in 2016, Grant Number LIFE15 IPE/IT/000013. Acknowledgments are given to all the Beneficiaries of the project LIFE-IP PREPAIR: Emilia-Romagna Region (Project Coordinator), Veneto Region, Lombardy Region, Piedmont Region, Friuli Venezia Giulia Region, Autonomous Province of Trento, Regional Agency for Environment of Emilia-Romagna, Regional Agency for Environment of Veneto, Regional Agency for Environment of Piedmont, Regional Agency for Environmental Protection of Lombardy, Environmental Protection Agency of Valle d'Aosta, Environmental Protection Agency of Friuli Venezia Giulia, Slovenian Environment Agency, Municipality of Bologna, Municipality of Milan, City of Turin, ART-ER, Lombardy Foundation for Environment

The forecasts for Kosovo were developed under the project: "Supply of project management, air quality information management, behaviour change and communication services" managed by the Millennium Foundation Kosovo (MFK), funded by the Millennium Challenge Corporation (MCC). Acknowledgments are given to all the Beneficiaries and Participants of the project for Kosovo, including Millennium Foundation Kosovo; the Hydrometeorological Institute of Kosovo and the National Institute of Public Health of Kosovo.

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
