# Peer review of "A standardized methodology for the validation of air quality forecast applications (F-MQO): Lessons learnt from its application across Europe"

_Geoscientific Model Development, 2023_

## Author Response (AR1)

*Dear Editor,*

*We thank the three anonymous referees for the constructive comments. It is worth to point out that Referees #2 and #3 agreed on their general evaluation of the paper, while Referee #1 had a different view and was more concerned about substantial concepts. Therefore, we tried to address the requests at our best, giving priority to Referees #2 and #3 in order to preserve the general coherence.*
*In the following, Referees' comment are reported in black text. Our point-by-point replies to each comment follow in blue italics. When we explicitly refer to changes in manuscript, blue bold italics is used.*

**Referee #1**

Review of "*A standardized methodology for the validation of air quality forecast applications (F-MQO): Lessons learnt from its application across Europe*" by Lina Vitali et al.

**Major comments:**

This paper presents a methodology for the validation of air quality forecast. The principle is based on the detection of sudden changes in pollutants concentrations, identification of threshold exceedances and capability to reproduce air quality indices. They apply their methodology to several forecasts across Europe. +The article is well-written and fairly long but contains problems of principle that make it unacceptable as is. These three major comments are described below. I suggest major revisions in order to give the authors the opportunity to improve the content of the study.

*We thank you for your review, which gave us the opportunity to improve the quality of the manuscript and to clarify some points. We addressed your suggestions at our best in the framework and the target of the present study. In some cases, your requests were beyond the scope of the paper and we explained the reason why we did not address them in full.*

1. The study suggests that the collaborative work will really blend the different approaches and draw out a general and new message. The confrontation of the different strengths and weaknesses could produce a more rigorous way of validating any type of forecast. But it is in fact a succession of applications of the same method on several particular cases. There is no homogeneous conclusion between the different results.

*The paper reports (cit.) "The application of the methodology to validate several forecasting simulations across Europe, using different modelling systems and covering various geographical contexts and spatial scales". It is indeed a collaborative work on different particular cases, but using one single validation approach.*

*We agree that the production of a more rigorous way of validating any type of forecast, after analyzing strengths and weaknesses of applications, is the final goal, and we wrote it in the conclusions ("further tests and analyses are needed in order to provide some criteria for defining the "fitness for purpose" of a forecasting application in predicting exceedances", "further developments are foreseen based on multi-category contingency tables and multi-categorical skill scores", "upgrades and improvements of the current validation protocol will be fostered by its usage"). Nevertheless, we believe that presenting the methodology and its results on specific applications is*

*an interesting first step to help identifying an adequate validation protocol in the context of the European Air Quality Directive.*

*Although comparing all results from all models, in an intercomparison approach, is not the scope of the paper, we tried to draw conclusions that are valid for all applications (e.g. FA1, FA2 and FA4 in Section 4.1). Multi-model conclusions are also discussed in the Conclusion section.*

***In the revised version of the manuscript, cross-cutting considerations were added based on additional analyses common to all applications (see the new Appendix C and Appendix E).***

2. The forecast qualification method (persistence model and the statistical scores) is really basic and contains nothing new. It is recommended to the authors to (i) propose new criteria to evaluate the forecasts,

*The proposed methodology relies on the combination of well-known statistical scores in the form of new composed statistical indicators (MQI, MPI) and simple-to-understand plots (Forecast Target Plot, Forecast MPI plot), which to our knowledge is not well covered in air quality literature. We see the fact, that these new indicators and plots are based on a well-known and accepted methodology used for evaluating modelled air quality assessments as an added value. So, concerning recommendation (i), the proposal of new criteria (i.e., $MQI_f$, $MPI_1$, $MPI_2$) and plots is already included within the proposed methodology. Further expanding the methodology is beyond the scope of the paper, which presents a framework and a tool for practical use. Our proposal to use the persistence benchmark in a systematic manner, including measurement uncertainty is also new to our knowledge.*

(ii) use the different forecasts and the different models to try to produce a more multi-model conclusion about the forecast and more independent of the case studied.

*Indeed, drawing up some multi-model conclusions is the main topic of the Conclusion section (see in particular lines 514-525 of the revised manuscript). Of course other tests on additional forecast applications covering other geographical areas (i.e. different meteorological and chemical regimes) will make these conclusions more robust. This is the reason why we concluded the paper fostering the usage of the proposed methodology by modelling developers and users in European Member States.*

The persistence model is too basic to provide information. The forecast must be validated by comparison to observations measurements with (D-1) (satellite, surface stations, soundings etc.). Comparing the forecast results to a persistence model that does not exist really, and is by principle erroneous, is a approach difficult to understand.

*The proposed evaluation is mainly based on the comparison with observations: nominator terms in $MQI_f$ and $MPI_x$ are RMSE and MFE (based on the model-observations differences) respectively. Persistence model is not used as the "truth" value but simply as a reference model, to be compared to the "truth" (i.e. the measurement value) as well. In other words, within the proposed protocol, two different aspects of model skills are included in a single metric: the agreement with measurements (nominator term) and the added value relative to a reference or baseline model (whose skills are at the denominator). Note that while the persistent model is indeed basic, it remains a challenge to over-perform it for most forecast models.*

*We added some sentences at lines 155-157 (tracked changes revised version of the manuscript) in Section 2.1 to clarify this point.*

3. I don't really agree with the expression 'fit for purpose' used several times. A chemistry-transport model is supposed to reproduce correctly the physics and chemistry independently of its use: analysis, scenario or forecast. It can have different scores depending on its chemical mechanism, resolution and therefore representativeness. There is no discussion on the representativeness of the models used and therefore no real interpretation on the score values according to the cases studied and the models.

*The concept of 'fitness for purpose' is widely used in literature (Hanna and Chang, 2012; Dennis et al., 2010; Baklanov et al., 2014) on modeling validation and in particular in the framework of the European initiatives FAIRMODE and HARMO (https://www.harmo.org/) both aiming at harmonizing atmospheric modelling usage and validation (Monteiro et al., 2018; Olesen, 1996).*

*Of course, a chemistry-transport model is supposed to reproduce correctly the physics and the chemistry of the atmosphere. Actually, checking the correctness of model algorithms, physics, assumptions, and codes is the first step of the evaluation of air quality models, i.e., scientific evaluation (Chang and Hanna, 2004). Once the model correctness is checked, an operational validation, based on statistical comparison with measurements is needed. Indeed, even if correct from both scientific and software points of view, no model can represent all the chemical-physical processes. Since it is an approximation of "real word" it is important to ensure that the level or complexity of the modelling is adequate for its intended use, in other words it is "fit-for-purpose". This is the aim of the operational validation, based on the comparison with measurements. Depending on the aim pursued, different evaluation strategies are put in practice (Baklanov et al., 2014). Moreover, specific criteria can be proposed (Boylan and Russell, 2006; Chemel et al., 2010) and, specifically, the defined criteria support the real interpretation on the score values. So, our approach, based on the concept of "fitness for purpose" is in line with literature on this topic.*

*We added citations in the manuscript to support the use of "fitness for purpose" concept.*

*Concerning the specific "purpose" addressed by the proposed evaluation protocol, we state in the introduction that "The main goal of a model evaluation process is to prove that the performances are satisfactory for its intended use, in other words, that it is "fit for purpose". Indeed, to be able to determine whether a model application is "fit for purpose", its purpose should be stated at the outset. Since air quality models are used to perform various tasks (e.g. assessment, forecasting, planning), depending on the aim pursued, different evaluation strategies should be put into practice." In the specific case of this study, the method aims to evaluate the capability of a modelling application (regardless of the type: CTM, lagrangian, machine learning etc.) to reach the "purpose" of producing air quality forecasts of practical use for short-term policy. The capability is evaluated quantitatively, by the numerical value of the $MQI_f$, in order to get a categorical judgment on the "fitness for purpose".*

*We rephrased where needed, to clarify the "purpose" of the applications concerned in this work.*

The conclusion is that additional tests are needed. Which ones and why not do them before submitting the work?

*We clarify that, as reported in the sections 4.3, 4.4 and 5, further tests are needed, but on one specific part of the protocol, i.e. the evaluation of forecast of exceedances. Section 4.3 already includes first tests of multi-category contingency tables on FA5. This is the direction of future work.*

*In general, a more extensive usage of the methodology in European Member States and beyond, especially for supporting policymaking (on additional forecast applications, covering other geographical areas and chemical-physical regimes), will reinforce the robustness of the approach and feedbacks will drive new developments. This is the reason why we made the methodology and the software publicly available.*

**Minor comments**

The introduction details well the different forecast platforms currently existing and their way to evaluate the quality of their forecast. Some sentences are very long and should be shortened. For example p.2, l.52

*Ok, done. **Some sentences were shortened.** Specifically concerning the point you are referring, at l.52 of p.2 of the original version of the manuscript (line 53 of the tracked changes revised version of the manuscript) a short sentence is proposed. We supposed you were referring to l.62 of p.2 of the original version of the manuscript (line 64 of the tracked changes revised version of the manuscript).*

Finally, the main goal of this paper is to add a new validation's methodology by calculating the model ability to simulate sudden changes and peaks of concentrations. First, it is not really new and second the use of the persistence model is not adapted to have realistic model vs observations comparisons.

*Concerning your first point ('it is not really new'), addressing Major Comment n.2 (i) we clarified that the novelty of the proposed methodology consists in the combination of well-known statistical scores in the form of new composed statistical indicators (i.e., $MQI_f$, $MPI_1$, $MPI_2$) and in the proposal for new simple-to-understand plots (Forecast Target Plot, Forecast MPI plot). The combination of these elements, and the inclusion of measurement uncertainty is new and its usage can provide more information than the sum of the single elements. Indeed, $MQI_f$, $MPI_1$, and $MPI_2$, by means of the new proposed plots, give new insights in model performances features, as discussed in Results and Discussion sections.*

*Concerning your second notice ('persistence model is not adapted to have realistic model vs observations comparisons') we agree. Addressing Major Comment n.2, we also clarified that the comparison is of course based on measurements data. Persistence model is not used for the comparison but as a benchmark. While it is a basic test (we indeed expect that model forecast beat the persistent model), this tests remains a challenge in practice.*

*We added sentences at lines 155-157 (tracked changes revised version of the manuscript) in Section 2.1 to clarify this point.*

2. Methodology

p.3 l.74: It is not always interesting to replace expression by acronyms. Too much acronyms tends to have a paper difficult to read. For example, you can explicitely write 'air quality forecast' in place of AQF.

*Thanks for the suggestion. **AQF acronym was replaced throughout the whole manuscript. AI and CTM acronyms were replaced as well**. Actually, only the acronyms AQI, AQD (widely used in literature) remain, along with the names of the statistical indicators, the models and the institutions.*

p.4 l.109: please define MQO acronym in its first use.

*Indeed, MQO was already defined in the original version of the manuscript (p.3 lines 84-85).*

2.1 It is questionable whether there is an interest to use the 'persistence model'. A lot of models and institutes are now doing forecast, then the use of such basic approach has not interest. Of course, to compare any other method to the 'persistence model' must provide better results. Statistically, the hazard is also probably better than the persistence model.

*Thanks for your comment which gives us the chance to make this topic clearer.*

*Concerning your prevision "compare any other method to the 'persistence model' must provide better results", **some additional outcomes were added in the revised version of the manuscript, in the framework of a sensitive analysis** suggested by Referee #3. The new Figure E1 in Appendix E shows that, also according to the current formulation (please refer to "1 Ur(RV)" outcomes), in some cases $MQO_f$ is not fulfilled. This is not an odd outcome: as we stated in the manuscript (lines 486 of the tracked changes revised version of the manuscript), according to Mittermaier (2008) "the difficulty in beating the persistence model skills is not infrequent".*

*Concerning your comment "Statistically, the hazard is also probably better than the persistence model", Mittermaier (2008) actually demonstrates the opposite, i.e. persistence model as a reference is better than the hazard one. Namely, Mittermaier (2008) shows the impact on the actual forecast performance evaluation of the choice of two different reference models, specifically the random forecast and the persisted forecast, the latter one being defined "the more realistic reference", as it "introduces the concept of "memory" to the idea of a reference forecast" and results in the end "far more realistic, but in being so, makes it harder for an actual forecast to beat".*

*Concerning the use of a real model as the reference model, an in-depth analysis was carried out according to anonymous Referee #3. Namely, CAMS ensemble model was used instead of persistence model and $MQI_f$ values were computed according to the new formulation. Outcomes showed that several issues would have to be addressed in order to make this option feasible. Globally, the outcomes of the tests were not robust enough to be included in the manuscript. More insights and in-depth analyses are needed which are beyond the scope of the current study.*

The whole section 2 does not provide any novelty: all statistical scores are already well known and published, widely used in all operational centers. Please highlight what is really new in this study.

*We tried to better explain the novelty of section 2 addressing Major Comment n.2 (i). Some points were reminded also above, addressing the second minor comment.*

3. Forecasting applications:

Table 2: The models are more than dispersion models, being chemistry-transport models.

*Ok, in Table 2 "Dispersion model" was changed in "Chemical Transport model", apart from FA3 (using Neural Networks approach) and FA5 (using the CALPUFF model).*

In general, the models presentation is very long and there is no comparison between the models (except the Table 2). To discuss the basis differences could provide more insight on the forecast results behaviour.

*Models presentation section was shortened in the revised version of the manuscript.*

*Concerning the comparison, indeed the forecast applications presented within this work are intentionally completely different: different models (and, concerning FA3, also different approach, i.e. Neural Networks), different domains and/or years (resulting in different meteorological and chemical regimes), different resolution, etc. The aim of the work is to find out if the proposed methodology can be applied in different contexts and to draw up some general conclusions. Discussing the behavior of each forecast application and comparing its features with others is not within the scope of the paper.*

4. Results, lessons learnt and discussion

p.11 l.280: There is an effort to try to categorize the results according to the scale of simulation, European, national, regional, but this does not really lead to a discussion of the forecasting issues related to these scales: boundary conditions, nesting, emission resolution etc.

*The paper does not present an intercomparison exercise. Indeed, forecast applications presented within this work are applied to different domains and refer to different years. So, discussing and comparing the different features of the forecast applications is not the scope of the paper. Actually, the paper is focused on the validation approach, investigating how it works in different geographical contexts. The idea is to find out if the same general and standardized evaluation method can be applied in different contexts and to draw up conclusions about the strengths and shortcomings of the approach, based on lessons learned from these different applications. At the point you are referring (p.11, l.280 of the original version of the manuscript), we specified this scope: "focusing on the lessons learnt by application of the proposed evaluation protocol to different geographical contexts and spatial scales and pointing out to strengths and shortcomings of the approach".*

p.13: Figures 1, 2, 3 and 4 are very complex, small and difficult to read. I was not able to read the 'summary statistics' part. And it is not really commented in the document.

*Thanks for your advice.*
**Figures 1, 2, 3, 4 were replaced by Figures 1-2, 3-4, 5-6, 7-8. Namely, in order to make the plots larger and clearer, we split each Figure (a panel of 4 plots in the original version) into 2 Figures (made of 2 plots in the revised version). Moreover, descriptive captions were included to each Figure in order to make them easier to be read.**

*References cited within this reply*

*Baklanov, A., Schlünzen, K., Suppan, P., Baldasano, J., Brunner, D., Aksoyoglu, S., Carmichael, G., Douros, J., Flemming, J., Forkel, R., Galmarini, S., Gauss, M., Grell, G., Hirtl, M., Joffre, S., Jorba, O., Kaas, E., Kaasik, M., Kallos, G., Kong, X., Korsholm, U., Kurganskiy, A., Kushta, J., Lohmann, U., Mahura, A., Manders-Groot, A., Maurizi, A., Moussiopoulos, N., Rao, S. T., Savage, N., Seigneur, C., Sokhi, R. S., Solazzo, E., Solomos, S., Sørensen, B., Tsegas, G., Vignati, E., Vogel, B., and Zhang, Y.: Online coupled*

regional meteorology chemistry models in Europe: current status and prospects, Atmospheric Chem. Phys., 14, 317–398, https://doi.org/10.5194/acp-14-317-2014, 2014.

Boylan, J. W. and Russell, A. G.: PM and light extinction model performance metrics, goals, and criteria for three-dimensional air quality models, Atmos. Environ., 40, 4946–4959, https://doi.org/10.1016/j.atmosenv.2005.09.087, 2006.

Chang, J. C. and Hanna, S. R.: Air quality model performance evaluation, Meteorol. Atmospheric Phys., 87, 167–196, https://doi.org/10.1007/s00703-003-0070-7, 2004.

Chemel, C., Sokhi, R. S., Yu, Y., Hayman, G. D., Vincent, K. J., Dore, A. J., Tang, Y. S., Prain, H. D., and Fisher, B. E. A.: Evaluation of a CMAQ simulation at high resolution over the UK for the calendar year 2003, Atmos. Environ., 44, 2927–2939, https://doi.org/10.1016/j.atmosenv.2010.03.029, 2010.

Dennis, R., Fox, T., Fuentes, M., Gilliland, A., Hanna, S., Hogrefe, C., Irwin, J., Rao, S. T., Scheffe, R., Schere, K., Steyn, D., and Venkatram, A.: A framework for evaluating regional-scale numerical photochemical modeling systems, Environ. Fluid Mech., 10, 471–489, https://doi.org/10.1007/s10652-009-9163-2, 2010.

Hanna, S. R. and Chang, J.: Setting Acceptance Criteria for Air Quality Models, in: Air Pollution Modeling and its Application XXI, Dordrecht, 479–484, https://doi.org/10.1007/978-94-007-1359-8_80, 2012.

Mittermaier, M. P.: The Potential Impact of Using Persistence as a Reference Forecast on Perceived Forecast Skill, Weather Forecast., 23, 1022–1031, https://doi.org/10.1175/2008WAF2007037.1, 2008.

Monteiro, A., Durka, P., Flandorfer, C., Georgieva, E., Guerreiro, C., Kushta, J., Malherbe, L., Maiheu, B., Miranda, A. I., Santos, G., Stocker, J., Trimpeneers, E., Tognet, F., Stortini, M., Wesseling, J., Janssen, S., and Thunis, P.: Strengths and weaknesses of the FAIRMODE benchmarking methodology for the evaluation of air quality models, Air Qual. Atmosphere Health, 11, 373–383, https://doi.org/10.1007/s11869-018-0554-8, 2018.

Olesen, H. R.: Toward the Establishment of a Common Framework for Model Evaluation, in: Air Pollution Modeling and Its Application XI, edited by: Gryning, S.-E. and Schiermeier, F. A., Springer US, Boston, MA, 519–528, https://doi.org/10.1007/978-1-4615-5841-5_54, 1996.

**Referee #2**

The paper presents an interesting extension of model validation approach based mainly to FAIRMODE activities and guidance documents.

Its novelty lies mainly in the extension of the already well documented methodology for forecast models. Also the case-studies documented and used in the paper are naturally interesting and also mostly new information.

*We thank you for your important suggestions and remarks, which gave us the opportunity to make the rationale of the work clearer.*

MAJOR

The paper states correctly that the criteria for model performance evaluation should be dependent on the purpose of the model but it should be stated even more clearly in the paper that the presented methodology is just an addon to the existing methodology,

The suggested metrics is probably useful for providing practical information specifically on the model forecasting capabilities,  but should be used ONLY together with the "standard" evaluation – not ever as a standalone measure of forecast models "fit-for-purpose" evaluation. Although the presented methodology seems to provide interesting and useful information for assessing forecast performance, but

it should be more clearly stated that the metric it defines is just one "random" choice: comparing everything to persistence model is easy to do  and provides some information on model forecast skills, but due to the extremely  simplistic nature of persistence model, the method does not necessarily capture all relevant measures for evaluating model performance

Authors should state   clearly, that this methodology (at least not yet at  this stage) does not really "validate" anything: there is not a chance to form any clear  and absolute criteria classifying model performance to acceptable/unacceptable  based on the presented methodology, so there is NOT enough  justification for recommending :

   "Therefore, it is recommended that forecasting applications fulfill the standard assessment MQO, as defined in Janssen and Thunis (2022), as well as the additional forecast objectives and criteria, as defined within the new specific protocol."

But, the methodology could be a good tool especially for model developers to find out forecast-specific issues in their models, especially  features which don' t show up in the standard statistical model evaluation . So my simple suggestion is, that the emphasis of the paper would be  clearly stated to be  more like  experimenting with different, forecast-relevant statistical metrics, but not  really claiming that the new index $MQOf$  would already be proven to be  good for serving as a real measure stating something definite on model forecasting skills or even comparing models with each other.

*It is important to note that the $MQO_f$ test is a necessary but NOT sufficient step for the forecast validation. This is why we recommend that both the assessment MQO and the forecast $MQO_f$ tests be fulfilled. Obviously, this does not mean that the validation process is completed, it only means that the first elementary tests have been performed. This is why the proposed approach adds other indicators and tests for completeness. We therefore do not see problems with this recommendation.*

*We agree that the methodology at this stage cannot be considered as a reference protocol to state that a model forecast is acceptable but it can be used to state that it is unacceptable.*

*We thank you for suggesting us to make these concepts clearer.*

*According to your advices in the revised manuscript version:*

✓ *we thoroughly rephrased the beginning of Section 2 in order to make the rationale of the proposed approach clearer. In particular, we explicitly stated that "The validation protocol proposed in this work … is an extension of the consolidated and well documented methodology fostered by FAIRMODE for the evaluation of model applications for regulatory air quality assessment", "it is recommended that the metrics suggested when evaluating forecasting applications are applied in addition to  the standard assessment MQO…", "Note that the proposed approach is not exhaustive. It does not evaluate all relevant features of a forecast application and other analyses will be helpful to gain further insights into the behaviour, the strengths and the shortcomings of a forecast application";*

✓ *at line 154  (of the tracked changes revised version of the manuscript), where $MQO_f$ is defined and proposed, we added some words in order to explicitly state that it refers to a specific skill (i.e. the capability to detect sudden changes of concentrations levels);*

✓ *throughout the whole manuscript we more clearly specify that the protocol, along with the defined criteria, are a proposal;*

✓ **We stressed the fact that the MQO_f should be seen as a necessary condition for use of forecasting results in a policy context but it is not a sufficient condition to ensure full quality. This is why additional metrics are proposed as part of the evaluation protocol.**

Instead, it would be good to add some discussions on the observed shortcomings of the methodology and suggest some improvements for it

*Some shortcomings of the methodology are described in the original version of the manuscript, concerning both the evaluation of the exceedances and the prediction of AQI levels. As far as AQI levels prediction is concerned, also the suggestion of some improvements and the direction of future work are presented (Section 4.3) along with the outcomes of the first tests of the application of the proposed upgrades.*

*On the contrary, we agree that shortcomings of the main MQO_f assessment were not sufficiently addressed. **In the revised version of the manuscript:***

✓ **we noted that "The magnitude of the MQIf score, since it is referenced to a benchmark, is dependent on the skill of the benchmark itself" (line 158 in Section 2.1 of the the revised version of the manuscript) and we explained how this issue is addressed within the proposed protocol;**

✓ **in Appendix E we evaluated to what extent changing measurement uncertainty parametrization changes performances evaluation outcomes.**

MINOR

1 One of the main justifications for the methodology seems to be paper by Mittermaier et al, 2008. , Please add some sentences including also the weaknesses of this NWP-persistence approach , showing clearly that also with the NWp's some issues were identified+ the main reason for not "officially" selecting some real model for reference seemed to be fear of failing too often against the reference, so the reason for suggesting persistence as reference was more a political compromise than choice justified by science.

*This is an important issue. Thank you very much for suggesting us to make this topic clearer.*
*Actually, Mittermaier (2008) shows the impact on the actual forecast performance evaluation ("Perceived Forecast Skill") of the choice of (only) two reference forecast, i.e. the random forecast and the persisted forecast, the latter one being defined "the more realistic reference". Indeed, Mittermaier (2008) states that random forecast is "generated to be completely independent of the observations, that is, pure chance". On the contrary, "unlike a random forecast, persistence introduces the concept of "memory" to the idea of a reference forecast". In Section 4, where the impact of reference forecast choice is illustrated, it is declared that "The visualization of a random forecast, such as the one shown in Fig. 2, highlights the lack of realism and resemblance of such a forecast to what is observed. In essence it represents the "kindest" reference for testing forecast skill in that it is one forecast we are never likely to make or observe! The other diagrams should make it clear that the persistent forecast is far more realistic, but in being so, makes it harder for an actual forecast to beat".*
*We did not find in the paper any reference to the use of some "real" model as the reference model.*
*Of course, it is true that also the weaknesses of the persistence approach was highlighted in Mittermaier (2008). In particular in the Concluding remarks it is stated that, persistence is a more realistic reference forecast, but it makes it harder to demonstrate the added value of an actual forecast, since the persistence is often difficult to beat. Moreover, the magnitude of a score, when referenced with respect to persistence, is strongly dependent on the skill of the persistence (reference) forecast (as it was demonstrated for both several hypothetical scenarios and real data from the*

*operational t+24 h Unified Model (UM) 12-km resolution regional forecasts). For this reason, Mittermaier (2008) suggests that "The skill in the reference forecast needs to be assessed and reviewed separately".*

*These are exactly the same arguments that suggested us to include the MPI Plot within the proposed protocol, in order to show the forecast performances regardless of the behavior of the persistence so supporting the interpretation of the outcomes, as discussed in section 4.2 of our paper.*

*We agree that it was not clearly stated when introducing MPIs metrics in Section 2.1.*

**We have added some explicit sentences at lines 155-158 of the tracked changes revised version of the manuscript.**

*Concerning the use of a "real" model as the reference model, an in-depth analysis was carried out according to anonymous Referee #3. Namely, CAMS ensemble model was used instead of persistence model and MQI$_f$ values were computed according to the new formulation. Outcomes showed that several issues would have to be addressed in order to make this option feasible. Globally, the outcomes of the tests were not robust enough to be included in the manuscript. More insights and in-depth analyses are needed which are beyond the scope of the current study.*

2 I might have just got lost in the very busy results section, but I did not find any reference/results for standard MQO-scores for the case-studies? This would help to understand how the model quality is related to model forecasting skill

*We appreciated this suggestion, improving the quality and the completeness of the global discussion.*

**Standard MQO outcomes for all available pollutants from all case-studies are now provided in Appendix C and used for supporting the discussion (e.g. at lines 366-368, 406-408, 427-428 of the tracked changes revised version of the manuscript).**

3 figures should in general be much clearer: now there is obviously lot of info presented in those, but in many cases , simply impossible to read

*Thanks for your suggestion.*

**Figures 1, 2, 3, 4 were replaced by Figures 1-2, 3-4, 5-6, 7-8. Namely, in order to make the plots larger and clearer, we spitted each Figure (a panel of 4 plots in the original version) into 2 Figures (made of 2 plots in the revised version).**

**Referee #3**

This study proposes a primary and two secondary Modelling Quality Indicators (MQI) to judge the performance of air quality forecast models against a primary Modelling Quality Objective (MQO) and secondary modelling performance criteria (MPC). The proposed method builds upon prior FAIRMODE work for objectively evaluating hindcast simulations and then applies the proposed method to five different datasets of air quality forecasts performed with different methods for different spatial domains. In terms of metric development, the new aspect relative to prior FAIRMODE work is the use of the persistence forecast as a benchmark. Like prior FAIRMODE work, the proposed metrics also account for what is termed "measurement uncertainty". After applying the proposed methods to assess the five forecast models, additional existing categorical and index-based evaluation metrics are applied to further assess the performance of these file models. The paper is well written and organized very clearly. The five models being used for illustration purposes represent a range of scales and methods.

*We appreciate the thorough review, with important suggestions and remarks. Addressing them improved the paper's completeness. Our responses to the two main concerns follow.*

My main concerns are that the specific primary MQI proposed here, and therefore the conclusion whether a model is or is not deemed to be "fit for purpose", a) rely on an intentionally uninformed forecast method (persistence) as benchmark, and b) are sensitive to the assumed and parameterized "measurement uncertainty", but the derivation and justification for this uncertainty is not included in this manuscript (it likely is included in earlier FAIRMODE publications). The "measurement uncertainties" shown in Figure A1 and affecting the calculation of the MQOs and Target Plots appear to be fairly large. For example, for fairly typical ozone concentrations around 120 ug/m3, the "measurement uncertainty" is about 20% (the y-axis value showing twice the uncertainty is about 0.4). I am guessing that "measurement uncertainty" refers not only to instrument error (which for ozone should be substantially lower than 20% for typical mean concentrations) but also to aspects like site representativeness relative to the grid, subgrid variability, and similar considerations. If so, "measurement uncertainty" doesn't seem to be the right term, and the value of that uncertainty in computing "allowable" model error should then also depend on model grid size since representativeness error relative to a grid is not an intrinsic station property. Again, this concept likely is well described in earlier FAIRMODE studies, but due to its important impact on the proposed forecast metrics, should also be explained and justified in the current study. If my guess is incorrect and this indeed is solely meant to represent instrument error, the values shown in Figure A1 appear to be unrealistically high (and therefore needlessly permissive of model error), at least for ozone.

*Thank you for suggesting us to make the 'measurement uncertainty' topic clearer and its formulation more documented.*

*Concerning measurement uncertainty documentation, we agree that "due to its important impact on the proposed forecast metrics, it should also be explained and justified in the current study".*
**We added details in both Section 2.1 (lines 145-149 of the tracked changes revised version of the manuscript) and Appendix A, where we also explicitly referred to the two key publications where uncertainty parameters are obtained for O₃, and NO₂ and PM respectively.**

*Concerning your comment about a rather high measurement uncertainty, first of all it should be mentioned that "Given that the final objective is to define a minimum level of performance to be fulfilled by air quality models, the following derivation will focus on estimating a "maximum measurement uncertainty"" (Thunis et al., 2013). We did not specify it in the original version of the manuscript and* **we clarified it in the revised version, where we added the following lines (lines 145-149 of the tracked changes revised version of the manuscript) to better explain the derivation of the measurement uncertainty:**
**"It is important to note that we use as representative for the measurement uncertainty the 95th percentile highest value among all uncertainty values. For PM10 and PM2.5 the results of a JRC instrument inter-comparison (Pernigotti et al. 2013) have been used whereas a set of EU AIRBASE stations available for a series of meteorological years has been used for NO₂ and analytical relationships have been used for O₃. These 95th percentile uncertainties only include the instrumental error."**
*As far as the Ozone uncertainty is concerned, its value around 120 ug/m3 is about 18%, i.e. very close to the data quality objective of 15% set as maximum allowed uncertainty for this pollutant in the 2008 air quality directives. Since the intention is to provide a test regarding the application of*

*models for policy purposes, it is important that the uncertainty values remain close to those set by legislation.*

*As stated in the few added lines, only the instrumental error is considered in the measurement uncertainty.*

I expect that both of these choices – an uninformed forecast method used as reference, and a rather high "measurement uncertainty" – led to the result that essentially all forecast models tested here meet the proposed MQO. It would be of interest to test how changing one or both of these aspects would affect the results. For example, given that CAMS routinely generates an 11 member ensemble forecast, one could use the CAMS ensemble mean forecast rather than persistence as a benchmark to assess whether any forecast models consistency outperform or underperform the ensemble, as indicated by the revised MQIf values no longer based on persistence as reference. Likewise, using lower estimates of "measurement uncertainty" (e.g. limiting it strictly to instrument error / precision) and then recalculating MQIf for the different forecast models might provide a more stringent test of model skill.

*We really appreciated this suggestion. We carried out some analyses according to both the aspects you suggested: a) how results are affected by using as a benchmark CAMS ensemble instead of persistence? b) how results are affected by lowering estimates of "measurement uncertainty"?*

*Concerning point b)* **The outcomes of the analysis were included in the revised version of the manuscript. Namely, they were provided in the new Appendix E** *improving the quality and the completeness of the global discussion.*

*Concerning point a) we agree that the persistence model has the shortcoming of being an intentionally uninformed forecast method and that using a more advanced model as benchmark could be interesting.*
*Anyway some issues would have to be addressed in order to make this option feasible.*
*First of all the choice of the model. No model can be considered as reference for every purpose and every scales. CAMS ensemble of course can be considered a good benchmark reference, if regional-scale background concentration are considered. At regional scale it is probably the best, being computed as an ensemble and being based on state-of-the-art European models. But this means that it could be very hard to beat it for other regional-scale models. On the contrary CAMS performances deteriorate at monitoring station with lower spatial representativeness, and when used as a benchmark in validation exercises of higher resolution forecast applications it could be beaten. Using CAMS ensemble as the same reference model could make the fitness for purpose criterion very difficult to beat at regional scale and easier at local scale. In other words, CAMS ensemble would not be impartial as a benchmark model.*
*The outcomes of the analysis provided here in Figure R1 confirm that. Figure R1 shows the results for all available data: FA2 and FA4 outcomes for the current forecast day (all pollutants available), and FA3 outcomes along a three-days forecast horizon (only PM10 available). FA1 was not considered because CAMS ensemble data are available for the download only for 3 years (FA1 refers to 2018).*
*Different colours refer to results based on different benchmark models. The results are provided by means of the violin plots, showing the distributions of the $MQI_f$ values computed for each monitoring station. Each violin refers to all the data provided within the corresponding Forecast Target Plot, giving in a single plot an overall view of all the outcomes available. Three lines were added to the display of each violin, indicating the $10^{th}$, the $50^{th}$ and $90^{th}$ percentiles of the distributions.*

*If CAMS ensemble is considered as the Benchmark model, FA2 and FA4 do not fulfil the $MQO_f$ criterion for any pollutants (the $90^{th}$ percentile of the $MQI_f$ values is always higher than 1) apart from FA2 for $O_3$. Also in cases where the criterion was not fulfilled also according to persistence (FA4 for $NO_2$ and PM2.5) using CAMS ensemble as the Benchmark model deteriorate the performance to a great extent.*

*When using CAMS ensemble as the Benchmark model, $MQO_f$ criterion remains fulfilled only for FA3 application, which not only has got very good performances, but also is based on learning on observations from the monitoring stations.*

*Another import issue arises from Figure R1: changing the Benchmark model turns out in changing not only $MQI_f$ values but also their distribution, highlighting a very different behaviour. This means that, if a new Reference Model was chosen, several new tests would have to be done in different contexts in order to understand how performances respond to the new approach.*

*In particular, from the above mentioned arguments it follows that a regional scale model could hardly beat CAMS ensemble according to the current criteria, i.e. $MQI_f \leq 1$ at more than 90% of the available stations. So, it should be necessary to define new criteria not impossible to be achievable for the common state-of-the-art forecast models. Furthered in-depth tests should be necessary to define the new criteria on the basis of several forecast applications carried out in different contexts.*

[Figure]

*Figure R1: Effect on the distribution of MQI$_f$ values of changing the Benchmark Model, form persistence to CAMS ensemble. Top: FA2, FA4 outcomes for the current forecast day (all the pollutants available). Bottom: FA3 outcomes along a three-days forecast horizon (only PM10 available).*

*Apart from the discussed arguments, another issue arises if a "real" model is used as a Benchmark model. This model could not be validated according to this protocol. A validation protocol targeting air quality forecasting services supporting policymaking in European Member States needs to be applicable to validate CAMS ensemble performances (it will be included in the incoming issue of the Daily Analyses and Forecasts EQC report, https://atmosphere.copernicus.eu/regional-services).*

*For all these reasons, even if we considered the suggested analysis very interesting and worth to be done, we think that it is not robust enough to be included in the manuscript. More insights and in-depth analyses are needed which are beyond the scope of the current study.*

Aside from these two major concerns, my only other comment is that I would have liked to see more in terms of diagnostic analyses, trying to use both the proposed MQIs and the application of the threshold-based and categorical metrics to gain further insights into the behavior of the different models. However, I realize that this may be outside the scope of the current study, but I hope that such work will be considered in the future.

*Thanks for the comment. We agree that it could have been interesting to better investigate into the behaviour of each presented forecast application, but it was outside the scope of the current study, focused on the validation approach more than on single forecast application performances. For this reason we focused on drawing up some general conclusions more than discussing single models behaviour.*
*What you suggested is of course of interest and* **we specifically add this purpose for future work, within Conclusions section (lines 532-534 of the tracked changes revised version of the manuscript).**